# Successful Dendrimer and Liposome-Based Strategies to Solubilize an Antiproliferative Pyrazole Otherwise Not Clinically Applicable

**DOI:** 10.3390/nano12020233

**Published:** 2022-01-11

**Authors:** Silvana Alfei, Andrea Spallarossa, Matteo Lusardi, Guendalina Zuccari

**Affiliations:** Department of Pharmacy, University of Genoa, Viale Cembrano, 16148 Genoa, Italy; andrea.spallarossa@unige.it (A.S.); matteo.lusardi@edu.unige.it (M.L.); zuccari@difar.unige.it (G.Z.)

**Keywords:** fifth-generation polyester-based lysine-modified dendrimer, physical encapsulation, 3-(4-chloro-phenyl)-5-(4-nitro-phenylamino)-1*H*-pyrazole-4-carbonitrile (CR232), water-soluble CR232-loaded dendrimer NPs, water-soluble CR232-loaded liposomes, spherical morphology, drug delivery systems, release profiles, release kinetics

## Abstract

Water-soluble formulations of the pyrazole derivative 3-(4-chlorophenyl)-5-(4-nitrophenylamino)-1*H*-pyrazole-4-carbonitrile (CR232), which were proven to have in vitro antiproliferative effects on different cancer cell lines, were prepared by two diverse nanotechnological approaches. Importantly, without using harmful organic solvents or additives potentially toxic to humans, CR232 was firstly entrapped in a biodegradable fifth-generation dendrimer containing lysine (G5K). CR232-G5K nanoparticles (CR232-G5K NPs) were obtained with high loading (DL%) and encapsulation efficiency (EE%), which showed a complex but quantitative release profile governed by Weibull kinetics. Secondly, starting from hydrogenated soy phosphatidylcholine and cholesterol, we prepared biocompatible CR232-loaded liposomes (CR232-SUVs), which displayed DL% and EE% values increasing with the increase in the lipids/CR232 ratio initially adopted and showed a constant prolonged release profile ruled by zero-order kinetics. When relevant, attenuated total reflectance Fourier transformed infrared spectroscopy (ATR-FTIR) and nuclear magnetic resonance (NMR) spectroscopy, scanning electron microscopy (SEM) and dynamic light scattering (DLS) experiments, as well as potentiometric titrations completed the characterization of the prepared NPs. CR232-G5K NPs were 2311-fold more water-soluble than the pristine CR232, and the CR232-SUVs with the highest DL% were 1764-fold more soluble than the untreated CR232, thus establishing the success of both our strategies.

## 1. Introduction

The diazole five-membered ring of pyrazole and its derivatives represent versatile template structures for designing new potent bioactive agents [1,2,3]. In fact, molecules belonging to the pyrazole family have numerous pharmacological activities, mainly attributable to the planar structure of the aromatic heterocycle [1,2,3]. According to Scopus, anti-inflammatory, analgesic, anticonvulsant, anthelmintic, antioxidant, and herbicidal effects of pyrazole have been reported since the year 1944 [4]. Several 3,5-diphenylpyrazole derivatives showed analgesic, hypotensive, anti-inflammatory, local anaesthetic, and motor activity inhibition effects in mice and rats, while mild platelet antiaggregating action in vitro [5]. On the contrary, the first studies reporting the antimicrobial effects and the cytotoxic action of pyrazole derivatives have been published more recently [6,7]. The naturally occurring amino acid *L*-α-amino-β-(pyrazolyl-*N*)-propanoic acid [(*S*)-β-pyrazolyl alanine], isolated in 1957 from the *Citrullus vulgarisin* juice of watermelon, was the first example of the pyrazole-containing natural product endowed with anti-diabetic activity [8].

Nowadays, the pyrazole nucleus is considered to have almost all types of pharmacological activities [9,10,11,12,13,14,15,16,17,18,19], and many researchers have studied this skeleton both chemically and biologically, reporting the synthesis and biological activity of pyrazole derivatives [2]. Functionalized pyrazoles and their fused analogues constitute the core structures of blockbuster drugs such as Viagra (against erectile dysfunction), Celecoxib (potent anti-inflammatory by COX-2 inhibition), Celebrex, Tepoxalin [nonsteroidal anti-inflammatory drugs (NSAIDs)], Crizotimib (anticancer), Acomplia, Surinabant, Difenamizole (anti-obesity), Mepiprazole (tranquillizer), Finopril (insecticide), CDPPB (antipsychotic), Betazole (analgesic), and Fezolamide (H2-receptor agonist and antidepressant agent), confirming the pharmacological potential of the pyrazole nucleus [2,3]. Within the anti-tumour therapeutic area, several pyrazole derivatives proved to be potent and selective inhibitors of protein kinases, a class of enzymes that, through the phosphorylation of different substrates, plays a pivotal role in the cell cycle [20,21]. Notably, the chemical studies that focused on both the development of novel procedures and the improvement of the existing protocols for the preparation and functionalization of pyrazoles are of outstanding interest in the field of medicinal chemistry.

Recently, through a novel synthetic strategy, a library of 29 pyrazole derivatives has been prepared and screened to assess their antiproliferative and cytotoxic activity against a panel of cancer cells and normal human fibroblasts, respectively [22]. Among the prepared pyrazole derivatives, compound CR232 (Figure 1) proved to significantly inhibit the growth of melanoma and cervical cancer cells [22]. Moreover, preliminary investigations have evidenced that CR232 could be promising also as a novel antibacterial agent, thus meeting the urgent global request for new therapeutic options against almost untreatable infections by bacteria increasingly resistant to available antibiotics [23,24,25]. Nevertheless, since it is water-insoluble and while the results from antimicrobial investigation are unclear, due to the tendency of CR232 to precipitate in the aqueous medium of the experiments, its future clinical application would remain utopic unless water-soluble CR232 formulations are developed.

Consequently, before performing further biological evaluations of CR232, such as dose- and time-dependent cytotoxicity experiments both on cancer cells and on normal ones, as well as before extending the antibacterial investigations to more bacteria of different species, we believe it is essential to make CR232 water soluble.

To solubilize drugs, without using harmful solvents such as di-methyl-sulfoxide (DMSO) or high amounts of surfactants and emulsifiers, often responsible for adverse reactions in patients, the most consolidated strategies consist of using nanosized reservoirs such as liposomes [26], hyperbranched polymers [27], star-like copolymer micelles [28] or dendrimers [29,30,31,32,33,34].

To our knowledge, except for one study recently published by us [33], only two studies exist in literature concerning the encapsulation of bioactive pyrazole derivatives in nanoparticles (NPs) [35,36] and only one regarding the application of nanotechnologies to enhance the water solubility of pyrazole [36]. Sun et al. [36] recently encapsulated the pyrazole derivative 6-amino-4-(2-hydroxyphenyl)-3-methyl-1,4-dihydropyrano [2,3-c] pyrazole-5-carbonitrile (AMDPC), found active as an anticancer agent, in poly (ethylene glycol) methyl ether-block-poly(lactide-co-glycolide) (PEG-PLGA), obtaining micelles, which gave clear water solutions at 0.05 mg/mL, while at the same concentration, pristine AMDPC was insoluble. Despite the authors success in improving the AMDPC water solubility, considering the very low DL% (DL = 1.28%) obtained in this study, the water solubility achieved for the encapsulated AMDPC was almost insignificant (6.4 × 10^−4^ mg/mL), thus making it not feasible or difficult for in vivo administration of the obtained AMDPC formulation without resorting to harmful solvents, such as DMSO.

To improve the results obtained by Sun, we thought that dendrimers could be of great help since they are three-dimensional macromolecules with a tree-like precise architecture and globular shape, very different from traditional polymers. Notably, peripherally cationic dendrimers of high generation are highly water-soluble due to the numerous peripheral hydrophilic groups compatible with water. On the other hand, they also have inner hydrophobic cavities capable of hosting a high number of lipophilic molecules, thus allowing us to obtain high DL% values and high water-solubility improvements.

In addition, since in the literature we found no study regarding the use of liposomes to enhance the water-solubility of bioactive pyrazole derivatives, we thought it could be interesting to explore this nanotechnological approach to improve the solubility profile of CR232. Specifically, only one study we found reported the encapsulation of 1-phenyl pyrazole-3, 5-diamine, 4-[2-(4-methylphenyl) diazenyl] and 1*H*-pyrazole-3 (1), 5-diamine, 4-[2-(4-methylphenyl) diazenyl] into liposomal chitosan emulsions for textile finishing [35].

According to not so recent articles, liposomes and specifically PEGylated “stealth liposomes” are mainly employed to reduce the proportion of ‘free drugs’ to avoid toxicity and to modify the pharmacokinetic and biodistribution profiles of drugs to improve their circulation time [37]. In this regard, several liposome-based formulations are currently used in the clinic for several applications [38]. It was recently reported that liposomal delivery systems have also drawn attention as one of the noteworthy approaches to increase the dissolution of water-insoluble drugs because of their biocompatibility and ability to encapsulate hydrophobic molecules in the lipid domain [39].

Liposomes, different from dendrimers that are derived from synthetic procedures, are biocompatible lipid NPs, not harmful for humans because of their natural origin, which, during the last decades have been extensively used in biomedicine, especially to transport and deliver antitumor drugs and antimicrobial agents [40]. Additionally, liposomes can protect the encapsulated drugs from environmental factors and early degradation, thus improving their performance features and therapeutic effects due to the reduced systemic toxicity [41,42]. Further, in addition to the need for fast production procedures, liposome-based drug formulations usually have reduced systemic toxicity [41,42].

In this study, we developed two different nanotechnological strategies to address the solubility downsides of CR232, which opposed both the feasibility of reliable microbiologic investigations and its possible future clinical applications, thus nullifying its potential as a therapeutic option. Particularly, while a highly hydrophilic lysine-containing fifth-generation biodegradable dendrimer synthetized by us (G5K) was firstly exploited to encapsulate and solubilize CR232, PEGylated liposomes made of natural lipids were secondly employed as encapsulating and solubilizing agents. In both cases, CR232-loaded NPs were obtained with enhanced water-solubility and properties, such as size, polydispersity indices, surface charge, DL%, EE%, and release profiles suitable for in vivo administration. Attenuated total reflection-Fourier transform infrared (ATR-FTIR) spectroscopy confirmed the success of both the encapsulation approaches, while nuclear magnetic resonance (NMR) analysis and potentiometric titrations confirmed the structure and completed the characterization of the cationic CR232-dendrimer formulation. Principal component analysis (PCA) was also exploited to process the ATR-FTIR spectral data of CR232, G5K, empty liposomes, CR232-loaded dendrimer NPs, and CR232-loaded liposomes, obtaining reliable predictive information about the chemical composition of all CR232 formulations prepared.

## 2. Materials and Methods

### 2.1. Chemical Substances and Instruments

The pyrazole derivative 3-(4-chloro-phenyl)-5-(4-nitro-phenylamino)-1*H*-pyrazole-4-carbonitrile (CR232) was prepared according to a procedure recently reported [22]. Since not reported in the previous work [22], the ATR-FTIR data of CR232 have been reported in Appendix A. Copies of the ATR-FTIR, ^1^H NMR, and ^13^C NMR spectra of CR232 have been included in Appendix A as Appendix A, Appendix A, and Appendix A, respectively.

The intermediate dendrimers necessary to prepare the fifth-generation cationic dendrimer (G5K), utilized for encapsulating the pyrazole derivative CR232, were prepared following a procedure reported in previous studies [43,44,45,46,47,48,49] and schematized in Appendix A available in Appendix A. Starting from *bis*-hydroxymethyl propanoic acid (*bis*-HMPA), which was used as an AB_2_-type monomer-building block, we first prepared the fourth generation polyester-based dendrimer with 48 peripheral hydroxyl groups (G4OH) [43,44,49]. Secondly, G4OH was furtherly esterified with 57.6 equivalents of *bis*-HMPA for the growth of one generation and to achieve G5OH [45,46,47]. The ATR-FTIR and NMR data, as well as the elemental analysis results of G4OH and G5OH, have been reported in Appendix A. If not differently specified, all chemical reagents including mannitol were purchased from Sigma Chemical Co. (St. Louis, MO, USA) or Merck (formerly Sigma–Aldrich, Darmstadt, Germany). They were reagent grade and were used without further purification. Hydrogenated soy phosphatidylcholine (HSPC), cholesterol (CHOL), and 1,2 distearoylglycero-3-phosphatidylethanolamine-*N*-polyethylene glycol-2000 (DSPE-PEG) were obtained from Avanti Polar Lipids, Inc. (Alabaster, AL, USA). Solvents were obtained from Merck (Darmstadt, Germany) and were purified by standard procedures. Melting points and boiling points were uncorrected. ^1^H and ^13^C NMR spectra of all compounds were acquired on a Jeol 400 MHz spectrometer (JEOL USA, Inc., Peabody, MA, USA) at 400 and 100 MHz, respectively. Fully decoupled ^13^C NMR spectra were reported. Chemical shifts were reported in ppm (parts per million) units relative to the internal standard tetramethylsilane (TMS = 0.00 ppm), and the splitting patterns were described as follows: s (singlet), d (doublet), t (triplet), q (quartet), m (multiplet), and br (broad signal). Centrifugations were performed on an ALC 4236-V1D centrifuge at 3400–3500 rpm. Elemental analyses were performed on an EA1110 Elemental Analyser (Fison Instruments Ltd., Farnborough, UK). Column chromatography was performed using Merck (Washington, DC, USA) silica gel (70–230 mesh) as a stationary phase. Scanning electron microscopy (SEM) images were obtained with a Leo Stereoscan 440 instrument (LEO Electron Microscopy Inc., Thornwood, New York, NY, USA). Dynamic Light Scattering (DLS) and Z-potential (ζ-p) determinations were performed using a Malvern Nano ZS90 light scattering apparatus (MalvernInstruments Ltd., Worcestershire, UK). Potentiometric titrations were performed with a Hanna Micro-processor Bench pH Meter (Hanna Instruments Italia srl, Ronchi di Villafranca Padovana, Padova, Italy). Lyophilization was performed using a freeze–dry system (Labconco, Kansas City, MI, USA). Thin layer chromatography (TLC) employed aluminium-backed silica gel plates (Merck DC-Alufolien Kieselgel 60 F254, Merck, Washington, DC, USA), and detection of spots was made by UV light (254 nm), using a Handheld UV Lamp, LW/SW, 6W, UVGL-58 (Science Company^®^, Lakewood, CO, USA). The organic solutions were dried over anhydrous magnesium sulphate. The removal of solvents was accomplished by using a rotatory evaporator (Rotavapor^®^ R-3000, Büchi Labortechnik, Flawil, Switzerland), operating at a reduced pressure of about 10–20 mmHg.

### 2.2. Experimental Procedures Concerning Dendrimer NPs

#### 2.2.1. Synthesis of the Fifth-Generation *^α^N*,*^ε^N*-(*tert*-Butoxycarbonyl)lysine Dendrimer (G5BK)

A solution of G5OH (72.6 mg, 0.00664 mmol) in dry DMF (1.5 mL) was added to *^α^N*,*^ε^N*-*bis*-(*tert*-butoxycarbonyl)lysine (di-Boc-Lys) (115.2 equiv., 265.2 mg, 0.7655 mmol), EDCI (115.2 equiv., 118.8 mg, 0.7655 mmol), and DMAP (57.6 equiv., 46.7 mg, 0.3825). The clear solution was kept under magnetic stirring at room temperature for 24 h. At the prefixed time, the mixture reaction was added with 20 mL ethyl acetate (EtOAc) to produce a suspension that was washed with 10% KHSO_4_ water solution (3 × 25 mL). The aqueous washings were extracted with EtOAc, and the combined organic phases were washed with 15% NaOH water solution, followed by water, then dried overnight (MgSO_4_). The removal of the solvent at reduced pressure afforded the functionalized dendrimer that had no need of further purification (207.5 mg, 0.004889 mmol).

Glassy solid (74% isolated yield). FTIR (KBr, ν, cm^−1^): 3380 (NH), 1747 (C=O ester), 1710 (C=O urethane), 1527 (NH). ^1^H NMR (CDCl_3_, 400 MHz): δ = 1.09–1.90 (m, 855 H, CH_3_ of G1, G2, G3, G4, G5) + CH_2_CH_2_CH_2_ of lys), 1.43 (s, 864 H, CH_3_ of Boc), 1.44 (s, 864 H, CH_3_ of Boc), 3.10 (m, 192 H, CH_2_NH), 4.25 (m, 474 H, CH_2_O of dendrimer + CHNH of lys), 4.70–5.50 (m, 192 H, ^α^NH + ^ε^NH). The CH_3_ of the core was not detected. ^13^C NMR (CDCl_3_, 100 MHz): δ = 14.20–17.90 (CH_3_ of G1, G2, G3, G4, G5), 22.57 (CH_2_), 28.36 (CH_3_ of Boc), 28.47 (CH_3_ of Boc), 29.57 (CH_2_), 31.84 (CH_2_), 40.04 (CH_2_NH), 46.42 (quaternary C), 53.37 (CHNH), 65.41–65.60 (CH_2_O of G1, G2, G3, G4, G5), 79.02 (quaternary C of Boc), 79.80 (quaternary C of Boc), 155.63 (C=O urethane), 156.17 (C=O urethane), 172.32 (C=O amino acid + C=O ester of G1, G2, G3, G4, G5), CH_3_, quaternary C, and CH_2_O of the core were not detected. Anal. Cald. for C_2006_H_3444_N_192_O_762_ requires C, 56.76; H, 8.18; N, 6.34%. Found: C, 56.41; H, 8.48; N, 6.33.

#### 2.2.2. Removal of *tert*-Butoxycarbonyl-Protecting Groups to Achieve G5K Hydrochloride Salt

A solution of the G5BK (207.5 mg, 0.004889 mmol) in 1 mL ethanol (EtOH) was cooled to 0 °C and treated with acetyl chloride (384.5 equiv., 133.5 µL, 1.88 mmol). The solution was kept at room temperature under magnetic stirring for 24 h and was then concentrated at reduced pressure, added to methanol (MeOH), and precipitated into acetone. The dendrimer in the form of hydrochloride salt was directly recovered as oil after centrifugation, washed repeatedly with fresh acetone, and regained by centrifugation at 3500 rpm for 15′. The obtained hygroscopic solid was dried at reduced pressure and stored under vacuum over P_2_O_5_.

Hygroscopic glassy solid, (97% isolated yield). FTIR (KBr, ν, cm^−1^): 3431 (NH_3_^+^), 1744 (C=O), 1635 (NH). ^1^H NMR (DMSO-*d6*, 400 MHz): δ = 1.03–1.40 (m, 279 H, CH_3_ of G1, G2, G3, G4, G5), 1.50–1.99 (m, 576 H, CH_2_CH_2_CH_2_ of lys), 2.76 (m, 192 H, CH_2_NH_3_^+^ of lys), 3.99 (m, 96 H, CHNH_3_^+^ of lys), 4.10–4.50 (m, 378 H, CH_2_O of dendrimer), 8.20 (br s, 288 H, NH_3_^+^), 8.82 (br s, 288 H, NH_3_^+^). CH_3_ of the core was not detected. ^13^C NMR (DMSO-*d6*, 100 MHz): δ = 19.33 (CH_3_), 23.14 (CH_2_), 28.01 (CH_2_), 31.01 (CH_2_), 40.02 (CH_2_NH_3_^+^), 47.70 (quaternary C), 53.55 (CHNH_3_^+^), 67.65–67.82 (CH_2_O and of G1, G2, G3, G4), 170.68–173.33 (C=O of amino acid + ester of G1, G2, G3, G4), CH_3_, quaternary C, and CH_2_O of the core were not detected. Anal. Cald. for C_1046_H_2100_N_192_O_378_Cl_192_ requires C, 41.57; H, 7.00; N, 8.90; Cl, 22.52%. Found: C, 41.88; H, 7.35; N, 9.15; Cl, 22.17.

#### 2.2.3. Determination of the Molecular Weight (MW) of Dendrimer G5K by Volumetric Titration

In addition to estimating the MW of the hydrochloride dendrimer G5K by its ^1^H NMR spectrum and having confirmed the obtained value by elemental analysis, we estimated it by volumetric titrations with HClO_4_ in acetic acid (AcOH), as previously reported by us for similar cationic dendrimers [44,45,48,49]. A sample of the dendrimer (10 mg) was dissolved in AcOH (5 mL) and treated with 2 mL of a solution of mercury acetate (1.5 g) in AcOH (25 mL); then, a few drops of a solution of quinaldine red (100 mg) in AcOH (25 mL) were added, followed by titration with a standardized 0.17–0.18 N solution of HClO_4_ in AcOH. The very sharp end point was detected by observing the disappearance of the red colour and the appearance of a fine white precipitate. Titrations were made in triplicate, and the result was reported as the mean ± standard deviation (SD).

#### 2.2.4. Cytotoxicity Studies on G5K

HeLa cell line, Dulbecco’s Modified Eagle Medium (DMEM), Fetal Bovine Serum (FBS, 10%), non-essential amino acids, antibiotics (penicillin and streptomycin), and 3-(4,5-dimethylthiazol-2-yl)-2,5-diphenyl-2*H*-tetrazolium bromide (MTT) were purchased from Termofischer Scientific (Rodano, Milan, Italy). Dose-dependent in vitro experiments were performed for the reservoir dendrimer G5K to investigate its cytotoxic effects on HeLa cells. Briefly, HeLa cells were increased in DMEM enriched with FBS, 10%, non-essential amino acids (1%), and antibiotics (1%, penicillin and streptomycin) and maintained in an atmosphere containing 5% CO_2_ at 37 °C. The cells were seeded at a density of 2 × 104 cells per well in a 24-well plate and in 4-wells slides in 500 µL of medium and incubated at 37 °C for 72 h. Subsequently, the cells were incubated with increasing concentrations (1–100 µM) of G5K at 37 °C for 24 h. Then, 10 µL MTT was added to each well and after 4 h, the medium and MTT were discarded and 100 µL DMSO was added to each well. Finally, optical density at 490 nm was measured on a Termofischer Scientific microplate reader (Rodano, Milan, Italy) to determine cell viability (%). Commercial fourth generation amine-terminated polyamidoamine dendrimer (G4-PAMAM-NH_2_) purchased from Merck (formerly Sigma–Aldrich, Darmstadt, Germany), having MW = 14214, was assayed under the same conditions as the positive control. Determinations were made in triplicate, and the results were expressed as the mean percentage of the control (untreated cells) ± SD.

#### 2.2.5. CR232-Loaded Dendrimer NPs (CR232-G5K NPs): Experimental Procedure

The CR232-G5K NPs were prepared by modifying and merging the widely reported solution casting method and the solvent diffusion evaporation method [50]. G5K dendrimer (Gen 5.0) was dissolved in 1 mL of MeOH (pH = 7.4). To the dendrimer methanol solution, a strong excess of CR232 (42.3 equiv.) and MeOH (1 mL) were added obtaining a suspension that was incubated for 3 h at 37 °C under vigorous stirring. Following incubation, the insoluble residue was separated, while the solution was evaporated. The evaporation of the alcoholic solution was performed at 70 °C under reduced pressure. The solid residue was suspended in 25 mL of water milli-Q, stirred for half an hour, and subsequently filtered. The residue was added to the first solid separated by the methanol solution and resuspended in methanol to dissolve all solutes. The solution was centrifugated (3500 rpm, 15′), separated by the residue, added to the water solution, and evaporated at reduced pressure. The thin film obtained was washed several times with acetone to remove residual water and subsequently with diethyl ether (Et_2_O), brought to constant weight under a vacuum, and stored in a dryer on P_2_O_5_ for further experiments (123.2 mg). The yellow washings containing the non-entrapped CR232 were added to the previously separated residues to collect all non-entrapped CR232 and were evaporated, obtaining a yellow powder. The recovered crude CR232 was recrystallized by DCM/petrol ether and was obtained as yellow crystals. Both ATR-FTIR analysis (not reported spectrum) and TLC (DCM/MeOH 9/1) confirmed the identity of the recovered solid as CR232.

#### 2.2.6. Spectroscopic Characterization of CR232-G5K NPs

ATR-FTIR (ν, cm^−1^): 3500–3000 (NH_3_^+^ dendrimer), 3331 (NH CR232), 2935 (alkyl groups of the dendrimer), 2226 (CN of CR232) 1735 (C=O stretching esters of dendrimer), 1602 (CH=CH stretching phenyl rings of CR232), 1508, 1376 (NO_2_ group of CR232) 1219, 1044 (C-O stretching esters of dendrimer).

^1^H NMR (DMSO-*d6*, 400 MHz): δ = < 1 (CH_3_
*core* not detected), 1.03–1.40 (m, 279 H, CH_3_ of G1, G2, G3, G4, G5), 1.50–1.99 (m, 576 H, CH_2_CH_2_CH_2_ of lys), 2.76 (m, 192 H, CH_2_NH_3_^+^ of lys), 3.99 (m, 96 H, CHNH_3_^+^ of lys), 4.10–4.50 (m, 378 H, CH_2_O of dendrimer), 7.61–7.67 (m, 82H, CH= aromatic ring), 7.68–7.70 (m, 82H, CH= aromatic ring), 7.84–7.86 (m, 82H, CH= aromatic ring), 8.16–8.18 (m, 82H, CH= of aromatic ring), 8.20 (br s, 288 H, NH_3_^+^), 8.82 (br s, 288 H, NH_3_^+^), 9.92 (bs, 41H, NH anyline, exchangeable with D_2_O), 13.87 (bs, 41H, H pyrazole, exchangeable with D_2_O). ^13^C NMR (DMSO-*d6*, 100 MHz): δ = 19.33 (CH_3_), 23.14 (CH_2_), 28.01 (CH_2_), 31.01 (CH_2_), 40.02 (CH_2_NH_3_^+^), 47.70 (quaternary C), 53.55 (CHNH_3_^+^), 67.65–67.82 (CH_2_O and of G1, G2, G3, G4), 84.35, 119.49, 120.62, 130.81, 133.51, 134.78, 140.38, 144.61, 153.86, 170.68–173.33 (C=O of amino acid + ester of G1, G2, G3, G4). CH_3_, quaternary C, and CH_2_O of the core of G5K, as well as expected signals at 152.20 and 157.15 of CR232 were not detected, and one signal of pyrazole was overlapped. From ^1^H NMR analysis: C_1702_H_2510_N_397_O_460_Cl_233_; MW = 44,153.1.

#### 2.2.7. UV-Vis Analyses

The UV-Vis spectra of CR232, CR232-G5K, and G5K were acquired using a UV-Vis spectrophotometer (HP 8453, Hewlett Packard, Palo Alto, CA, USA) equipped with a 3 mL cuvette. With the use of a microbalance, 4.988 mg of CR232 and 6.000 mg of CR232-G5K were solubilized in DMSO (50 mL), obtaining solutions with concentrations of 99.76 µg/mL and 120 µg/mL, respectively. Then, 48.5 mg of G5K was solubilized in 1 mL DMSO, obtaining a mother solution at a concentration 48.5 mg/mL; 100 µL of the obtained solution was diluted (1/10), obtaining a solution with a concentration 485 µg/mL. The ultraviolet absorption spectra of CR232, CR232-G5K, and G5K were recorded in the range 230–650 nm.

#### 2.2.8. CR232 Calibration Curve

A stock solution of CR232 (0.1 mg/mL) was prepared in DMSO by diluting an initial solution obtained by dissolving 5 mg of CR232 in 50 mL of DMSO. Dilutions with DMSO were made to prepare standard solutions at concentrations of 0.02494, 0.01995, 0.01496, 0.00998, and 0.00499 mg/mL. The content of CR232 present in each solution was quantified using the UV-Vis apparatus described in the previous Section 2.2.7 by detecting the absorbance ([A]) at room temperature and ʎ_abs_ = 384 nm. The CR232 calibration curve was obtained by the least-squares linear regression analysis of the CR232 concentrations vs. the [A] signals created in the UV detector by the different concentrations of the analyte (CR232). Determinations were made in triplicate, and the [A] values obtained for each CR232 concentration analysed were expressed as the mean ± SD. The equation of the developed linear calibration model was the following Equation (1).
y = 60.019x + 0.0107 (1)
where y is the [A] values measured at ʎ_abs_ = 384 nm, and x the CR232 standard concentrations analysed. In Equation (1), the slope represents the coefficient of extinction (ε) of CR232.

#### 2.2.9. Estimation of CR232 Contained in CR232-G5K NPs

Here, 6.0 mg of CR232-G5K was dissolved in 50 mL of DMSO, yielding a 0.120 mg/mL solution, which was analysed as such and then further diluted to obtain CR232-G5K clear solutions at different concentrations (60, 30, 24, and 12 µg/mL), which were vigorously stirred for ten minutes to promote the release of CR232. The amount of CR232 in the samples of CR232-G5K NPs was quantified by UV-Vis analysis using the UV-Vis apparatus previously described and Equation (1). The samples of CR232-G5K NPs were analysed against a blank solution of the empty dendrimer G5K. Determinations were made in triplicate, and the results were expressed as the mean of three independent experiments ± SD. After estimating the content of CR232 in CR232-G5K NPs, the DL% and EE% values of CR232-G5K NPs were calculated from the following Equations (2) and (3).
(2)$$\mathrm{DL}\;\left(\%\right)=\frac{\mathrm{weight}\;\mathrm{of}\;\mathrm{the}\;\mathrm{drug}\;\mathrm{in}\;\mathrm{NPs}}{\mathrm{weight}\;\mathrm{of}\;\mathrm{the}\;\mathrm{NPs}}\times 100$$
(3)$$\mathrm{EE}\;\left(\%\right)=\frac{\mathrm{weight}\;\mathrm{of}\;\mathrm{the}\;\mathrm{drug}\;\mathrm{in}\;\mathrm{NPs}}{\mathrm{inizial}\;\mathrm{amount}\;\mathrm{of}\;\mathrm{drug}}\times 100$$

#### 2.2.10. Molecular Weight of CR232-G5K NPs

According to a previously reported procedure [32,33], the MW of CR232-G5K NPs was estimated both by ^1^H NMR analysis and by data obtained from UV-Vis analyses, which provided the estimate of the moles of CR232 loaded per dendrimer mole.

#### 2.2.11. Morphology and Average Size of G5K and CR232-G5K NPs

The morphology and average size of G5K and of CR232-G5K NPs were investigated by scanning electron microscopy (SEM) analysis. In the performed experiments, samples were fixed on aluminum pin stubs and sputter-coated with a gold layer of 30 mA for 1 min, and an accelerating voltage of 20 kV was used for the sample’s examination. The micrographs were recorded digitally using a DISS 5 digital image acquisition system (Point Electronic GmbH, Halle, Germany).

#### 2.2.12. Potentiometric Titrations of CR232-G5K NPs

Potentiometric titrations were performed at room temperature to construct the titration curves of G5K and CR232-G5K NPs and obtain the corresponding first derivatives. The samples (20–30 mg) were dissolved in 30 mL of Milli-Q water (m-Q) and treated with a standard 0.1 N NaOH aqueous solution [1.5 mL, pH = 10.20 (G5K) and 10.50 (CR232-G5K)]. The solutions were potentiometrically titrated by adding 0.2 mL aliquots of a standard 0.1 N HCl aqueous solution, up to a total of 3.0 mL, followed by measurements of the corresponding pH values [33,44,45,48,49,51]. Titrations were made in triplicate, and the determinations were reported as the mean ± SD.

#### 2.2.13. In Vitro CR232 Release Profile from CR232-G5K NPs

The release profile of CR232 from CR232-G5K NPs was investigated in vitro using the dialysis bag diffusion method. Notably, 5.0 mg of NPs exactly weighted was dissolved in 2 mL of 0.1 M phosphate-buffered saline (PBS, pH = 7.4), which should assure the dissolution of the complex according to the results obtained from water solubility determinations. Additionally, corresponding to the determined DL%, the sample should contain 1.583 mg of CR232. The solution was then positioned in a pre-swelled T2 tubular cellulose dialysis bag (flat width = 10 mm, wall thickness = 28 µm, V/cm = 0.32 mL, Membrane Filtration Products, Inc., Seguin, TX, USA) with a nominal molecular weight cut-off (MWCO) of 6000–8000 Da, bathed in 20 mL of 0.1 M PBS (pH 7.4, 37 °C), and gently stirred for 24 h. At predetermined time intervals (0 h, 1 h, 2 h, 3 h, 4 h, 5 h, 6 h, 8 h, 10 h, 12 h, 24 h), 10 mL was withdrawn from the incubation medium, evaporated at reduced pressure, and brought to constant weight at high vacuum, obtaining ten yellow solid residues. Then, the residues were re-dissolved in DMSO, opportunely diluted with the same solvent to allow the quantification of the exact amount of CR232 present in the samples by using the UV-Vis apparatus described in Section 2.2.7 and Equation (1). The [A] measurements were made at 384 nm in triplicate, and the results were reported as the mean ± SD of three determinations. After sampling, an equal volume of fresh PBS was immediately replaced in the incubation medium.

The concentrations of CR232 released from CR232-G5K NPs were expressed as a cumulative release percentage (CR%) of the total amount of CR232 loaded in the CR232-G4K NPs in respect of the DL% value determined. To obtain the CR232 release profile from the dendrimer-based NPs, the CR232 CR% were plotted vs. times in a dispersion graph.

### 2.3. Experimental Procedures Concerning Liposomes

#### 2.3.1. CR232-Loaded Unilamellar Liposomes (CR232-SUVs): Experimental Procedure

Initially, multilamellar vesicles (MLVs) were prepared according to the film hydration method using a standard initial molar ratio of lipids 2/1 [52]. Briefly, aliquots of 10 mM HSPC, CHOL, and DSPE-PEG chloroform stock solutions were mixed to yield a HSPC/CHOL/DSPE-PEG 2/1/0.1 molar ratio, with a fixed total lipid amount equal to 0.03 mmol. CR232 was dissolved in tetrahydrofuran (THF) and then was combined with lipids at various CR232 concentrations (0.4, 0.8, 2.0 mg/mL), corresponding to a total lipids/CR232 ratio of 5/1, 15/1, and 30/1, and to a chloroform/THF ratio of 3/1, 7.5/1, and 15/1. The thin films at different ratio lipids/CR232 were obtained by roto-evaporating the organic solvents under vacuum at 40 °C for 1 h to ensure the total removal of solvents. The films were hydrated at a temperature over the gel–liquid transition temperature of the amphiphiles (Tc) with 4 mL of distilled water. Following hydration, liposomes were extruded (LiposoFast-basic extruder; Avestin Inc., Ottawa, Canada) through series of polycarbonate filters of pore size ranging from 400 nm to 100 nm, thus leading to translucent suspensions of small unilamellar vesicles (SUVs), here named CR232-SUVs. CR232-SUVs suspensions were used as such for determining the related CR232 concentration, the EE%, and their water solubility. In parallel, following the same procedure, SUVs without CR232 were prepared as blanks. To perform ATR-FTIR, all prepared water SUV suspensions were freeze-dried without cryoprotectant. Briefly, proper volumes of SUVs and of CR232-SUVs 5/1, 15/1, and 30/1 (ratios lipids/CR232) suspensions were aliquoted in 5 mL glass vials and first frozen at −20 °C. Subsequently, they were placed into the lyophilization chamber set at −30 °C. Sublimation occurred by reducing the pressure to less than 20 × 10^−3^ mbar (Labconco, Kansas City, MI, USA). After 48 h, secondary drying was performed by increasing the temperature to 25 °C for 1 h. The freeze-dried micelles were stored at 4 °C until further use.

In parallel, the same SUV water suspensions were mixed with trehalose or sucrose, as cryoprotectants, at a 1/10 lipids/cryoprotectant molar ratio [52]. Aliquots of cryoprotectant 0.1 M stock solutions (0.3 mmol) were added to the freshly prepared liposomes, and the obtained cryoprotectant-added liposome suspensions were treated as previously described for those without cryoprotectant. The freeze-dried liposomes obtained were immediately observed, compared among each other, and compared with the freeze-dried liposomes obtained without using any cryoprotectant to evaluate the efficiency of the two cryoprotectants employed. Subsequently, they were stored at 4 °C until further use.

#### 2.3.2. Spectroscopic Characterization of SUVs and of CR232-SUVs

SUVs ATR-FTIR (ν, cm^−1^): 3600–3000 (NH, OH stretching of phospholipids and cholesterol), 2917, 2850 (alkyl groups of fatty acids), 1736 (C=O stretching esters of phospholipids), 1467, 1377 (CH_2_ and CH_3_ banding), 1235 [R–PO_2_–R′ (diester phosphate)], 1176 (C-O stretching esters).

CR232-SUVs 5/1 ATR-FTIR (ν, cm^−1^): 3600–3000 (NH, OH stretching of phospholipids and cholesterol), 2917, 2850 (alkyl groups of fatty acids), 1736 (C=O stretching esters of phospholipids), 1467, 1377 (CH_2_ and CH_3_ banding), 1236 [R–PO_2_–R′ (diester phosphate)], 1175 (C-O stretching esters).

CR232-SUVs 15/1 ATR-FTIR (ν, cm^−1^): 3600–3000 (NH, OH stretching of phospholipids and cholesterol), 3333 (NH of pyrazole), 2917, 2850 (alkyl groups of fatty acids), 2225 (CN of pyrazole), 1736 (C=O stretching esters of phospholipids), 1604 (CH=CH stretching phenyl rings of pyrazole), 1486 (NO_2_ group of pyrazole), 1467, 1377 (CH_2_ and CH_3_ banding), 1336 (NO_2_ group of pyrazole), 1236 [R–PO_2_–R′ (diester phosphate)], 1176 (C-O stretching esters).

CR232-SUVs 30/1 ATR-FTIR (ν, cm^−1^): 3600–3000 (NH, OH stretching of phospholipids and cholesterol), 2917, 2850 (alkyl groups of fatty acids), 2226 (CN of pyrazole), 1736 (C=O stretching esters of phospholipids), 1486 (NO_2_ group of pyrazole), 1467, 1378 (CH_2_ and CH_3_ banding), 1336 (NO_2_ group of pyrazole), 1235 [R–PO_2_–R′ (diester phosphate)], 1175 (C-O stretching esters).

#### 2.3.3. Estimation of CR232 Contained in CR232-SUVs

The CR232-SUVs suspensions at different ratios of lipids/CR232 were purified from non-incorporated CR232 by gel chromatography on a Sephadex G50 [52]. Since the gel chromatography isolation process is based on MW, liposomes passed more freely than CR232 due to size restriction. On the contrary, free CR232 entered the pores in the bead and was eluted by the mobile solvent after liposomes. After elution of the column void volume, we obtained five opalescent fractions for each original suspension that together had a mean volume of about 3 mL. The so purified suspensions were diluted in DMSO (500 µL of liposome suspensions was diluted to 3 mL with DMSO), and liposomes were disrupted by sonication for 10 min to allow the release of CR232. The drug content, which corresponded to the water solubility of CR232 present in the vescicles, was quantified as described for CR232-G5K NPs. Determinations were made in triplicate, and the results were expressed as the mean of three independent experiments ± SD. Then, the EE% value was determined using Equation (4).
(4)$$\mathrm{EE}\;\left(\%\right)=\frac{\mathrm{weight}\;\mathrm{of}\;\mathrm{the}\;\mathrm{drug}\;\mathrm{in}\;\mathrm{SUVs}}{\mathrm{inizial}\;\mathrm{feeding}\;\mathrm{drug}}\times 100$$

The DL% of CR232-liposome formulations prepared with diferent ratios of lipids/CR232 was determined on the freeze-dried samples according to the formula in Equation (5).
(5)$$\mathrm{DL}\;\left(\%\right)=\frac{\mathrm{weight}\;\mathrm{of}\;\mathrm{the}\;\mathrm{drug}\;\mathrm{in}\;\mathrm{SUVs}}{\mathrm{weight}\;\mathrm{of}\;\mathrm{the}\;\mathrm{SUVs}}\times 100$$

Briefly, a weighed amount of the lyophilized SUV powders was solubilized in DMSO, and the CR232 content was spectrophotometrically assessed by UV-Vis analyses. The lipid concentration was derived by subtracting the CR232 and the cryoprotectant contribution to the weighed samples. Determinations were made in triplicate, and the results were expressed as the mean of three independent experiments ± SD.

#### 2.3.4. In Vitro CR232 Release Profile from CR232-SUVs

Release studies of CR232 from CR232-SUVs were carried out at 37 °C for 24 h directly after lyophilization. Briefly, lyophilized void and loaded liposome samples (drug content = 2 mg) were reconstituted with 1 mL of PBS buffer (pH 7.4) and placed in a dialysis bag (Spectra/Por Float-A-Lyser G2, CE, m.w. cutoff = 100 kDa), hermetically sealed, and immersed in 20 mL of PBS with 0.5% Tween 80 to ensure sink conditions.

The entire system was kept in a thermal bath with constant stirring; 2 mL aliquots of the dialysate were withdrawn from the receptor medium at predetermined time intervals (3, 6, 16, 20, and 24 h) and replaced with fresh medium. The samples were subsequently diluted with DMSO before being transferred to the UV-Vis spectrophotometer for the determination of the CR232 content, as described in the previous section. Liposomes without CR232 were used as blanks. Determinations were made in triplicate, and the results were expressed as the mean of three independent experiments ± SD. The CR232 cumulative release percentage (CR%) was calculated by dividing the cumulative amount of the CR232 recovered in the dialysis medium by the total weight of CR232 present in the liposomes. To obtain the CR232 release profile from the liposome-based NPs, the CR232 CR% was plotted vs. time in a dispersion graph.

### 2.4. Analytical Experiments Concerning Both Dendrimer NPs and Liposomes

#### 2.4.1. Principal Component Analysis (PCA) of ATR-FTIR Spectral Data

ATR-FTIR spectra of CR232-G5K, CR232-SUVs, CR232, SUVs, and G5K were recorded in triplicate on a Spectrum Two FT-IR Spectrometer (PerkinElmer, Inc., Waltham, MA, USA). Acquisitions were made from 4000 to 600 cm^−1^, with a 1 cm^−1^ spectral resolution, co-adding 32 interferograms, with a measurement accuracy of the frequency data at each measured point of 0.01 cm^−1^ due to the laser internal reference of the instrument. The “find peaks” tool of the instrument software was used to automatically obtain the frequency of the main bands. The FTIR data sets of the acquired spectra were organized in matrices 3001 × *n* of measurable variables, where *n* is the number of samples. For each sample, the variables consisted of the values of transmittance (%) associated with the wavenumbers (3001) in the range 4000–1000 cm^−1^.

Three matrices of spectral data were created. The first was a matrix 3001 × 3 (9003) collecting data of CR232-G5K, CR232, and G5K, the second was a matrix 3001 × 5 (15005) made of data of the three CR232-SUVs formulations (5/1, 15/1 and 30/1), CR232, and SUVs, and the last one was a matrix 3001 × 7 (21007) collecting the spectral data of all samples. Each matrix was subjected to PCA using PAST statistical software (paleontological statistics software package for education and data analysis, free down-loadable online, at: https://past.en.lo4d.com/windows; accessed on 28 December 2021).

#### 2.4.2. Water Solubility Studies

The water solubility of the untreated CR232 pyrazole derivative, of CR232-G5K NPs, and of nano-manipulated free CR232 contained in G5K NPs was determined by the shake-flask method [33,50,53]. Excesses of CR232 (4.5 mg) and of CR232-G5K (6.1 mg) were added to water m-Q (2 mL and 1 mL, respectively), obtaining suspensions that were incubated at 37 °C and stirred vigorously, observing (mainly for the free CR232) abundant foaming (pH = 7.4). The suspensions were maintained under stirring to promote the achievement of an equilibrium between the saturated solutions and the undissolved samples. Then, the suspensions were centrifugated (15 min, 3350 rpm) to precipitate the non-solubilized material, which was separated by the supernatant solutions. The pale-yellow solid residues were washed several times with acetone to eliminate the residual water and brought to constant weight at reduced pressure. The supernatant solutions obtained were filtered using a 0.22 µm filter, and after having observed drops of the solutions with a Leica Galen III Professional Microscopes (Taylor Scientific, St. Louis, MO, USA) without detecting precipitate or differences with a drop of pure water, they were evaporated at reduced pressure and brought to constant weight under high vacuum, obtaining fractions of samples that were water soluble (0.0045 ± 0.0002 mg for CR232 and 5.2 ± 0.05 mg in the case of CR232-G5K NPs). The amounts of the separated insoluble residues were 4.50 ± 0.03 mg for CR232 and 0.91 ± 0.02 mg for the CR232-G5K NPs, thus confirming the reliability of the weights obtained for the soluble solids, with errors of 0.1% and 1%, respectively. The experiments were performed in triplicate, and the solubilities of untreated CR232, of CR232-G5K NPs, and of CR232 contained in G5K NPs were reported as the mean ± SD. The water solubility of CR232 contained in the CR232-SUVs was determined directly on the liposome water suspensions obtained after hydration of the lipid films 5/1, 15/1, and 30/1 with distilled water. Briefly, after extrusion and purification by gel chromatography on Sephadex G50, aliquots of each suspension were diluted with DMSO and analyzed by the UV-Vis apparatus previously described. The values of [A] detected at ʎ_abs_ = 384 nm and at room temperature were used to determine the CR232 concentrations (mg/mL) by using Equation (1), which corresponded to the water-solubility of CR232 contained in the liposome-based formulations. Additionally, the lyophilized CR232-SUVs 30/1 formulation that resulted in the highest EE% and DL% was used to determine the water solubility of the CR232-loaded liposomes by the shake-flask method described above. All determinations on liposomes were made in triplicate, and the solubilities were reported as the mean ± SD.

#### 2.4.3. Dynamic Light Scattering (DLS) Analysis

The particle size (in nm), polydispersity index (PDI), and zeta potential (ζ-p) (mV) of G5K, CR232-G5K NPs, and CR232-SUVs were measured at 25 °C, at a scattering angle of 90° in m-Q water by using a Malvern Nano ZS90 light scattering apparatus (Malvern Instruments Ltd., Worcestershire, UK). Solutions of samples in m-Q water were diluted to final concentrations to have 250–600 kcps. The ζ-p value of all samples was recorded with the same apparatus. The results from these experiments were presented as the mean of three different determinations ± SD. Concerning the particle size distribution, intensity-based results were reported.

### 2.5. Statistical Analysis

The statistical significance of the slope of the CR232 calibration curve was investigated through analysis of variance (ANOVA), performing Fisher’s test. Statistical significance was established at a *p*-value < 0.05.

## 3. Results and Discussion

### 3.1. Dendrimer NPs

#### 3.1.1. Synthesis of Boc-Protected Dendrimer G5BK Containing Lysine Residues

To prepare the lysine containing the fifth-generation dendrimer, different from the procedure previously reported [44], we performed a simpler and fast procedure, which started from the uncharged dendrimer G5OH. Briefly, the proper number of Boc-protected lysine equivalents was directly grafted onto G5OH by an esterification reaction utilizing the coupled 1-ethyl-3-(3-dimethylaminopropyl) carbodiimide (EDC) and 4-dimethylaminopyridine (DMAP) as a coupling agent and catalyst, respectively, in dimethylformamide (DMF) as a solvent, for 24 h at room temperature (Figure 1).

The use of basic EDC allowed us to easily remove the side products including the ureic and acylureic by-products derived from EDC by acid washings and extractive work-up after hydrolysis and to obtain analytically pure G5- BK, with no further purification.

G5BK is a glassy solid, soluble in almost all organic solvents except for pentane, hexane, cyclohexane, petroleum ether, and diethyl ether. The ATR-FTIR and NMR spectral data, as well as the elemental analysis results, were like those obtained for the same molecule prepared following a different synthetic procedure [44] and confirmed its structure.

#### 3.1.2. Removal of *tert*-Butoxycarbonyl-Protecting Groups to Achieve G5K Hydrochloride Salt

The successive removal of Boc groups to achieve the cationic dendrimer G5K as a hydrochloride salt was performed with anhydrous HCl produced in situ by reacting acetyl chloride with ethanol, conditions that proved to be compatible with the ester matrix of the dendrimer (Figure 1). G5K was obtained as highly hygroscopic glassy solid that was stored under vacuum over P_2_O_5_. The ATR-FTIR and NMR spectral data confirmed the structure of G5K, and copies of the spectra are available in Appendix A. In this study, the structure of G5K and its MW, estimated both by volumetric titrations and by its ^1^H NMR spectrum, were also confirmed by elemental analysis.

#### 3.1.3. Determination of the MW of Dendrimer G5K by Volumetric Titration

To determine the MW of G5K and to have additional evidence of its structure and peripheral composition, the established and previously validated [44,45,48,49] technique consisting of the titration of the amine hydrochloride groups with HClO_4_ solutions in AcOH in the presence of mercuric acetate and quinaldine red as an indicator [54] was performed. Note that we have been the pioneers of using this innovative, cheap, and fast method for cationic dendrimers. Its accuracy is secured by a sharp endpoint of titration, while its reliability has been demonstrated by the reproducibility of results. Table 1 collects the comparison (including the percentage error) between the MW of G5K estimated by the ^1^H NMR and is confirmed by results of elemental analysis and the MW obtained by volumetric titrations and proper calculations.

The good agreement (error < 5%) of the MW obtained by volumetric titrations with that estimated by the G5K ^1^H NMR spectrum confirmed the molecular structure of the prepared dendrimer and the suitability of the method.

#### 3.1.4. Cytotoxicity Studies

Since water solubility is a crucial requirement that bioactive compounds should have to be administrable and efficacious in vivo, the main scope of the present study was to develop successful strategies based on nanotechnology to enhance the insignificant water solubility of CR232. Experiments in animals suggested that NPs might be toxic to humans. Moreover, epidemiological studies suggested the existence of a relationship between NP pollution and human diseases, and the observation of the presence of NPs within diseased human tissue enforced these ideas [55]. The intrinsic cytotoxicity of G5K NPs should be considered in future in vivo evaluation of CR232-G5K NPs to avoid unpleasant collateral effects due to the presence of the nanosized carrier. Dose-dependent cytotoxicity experiments for G5K were performed in vitro using HeLa cells and the MTT assay. In parallel, a commercial G4-PAMAM-NH_2_ dendrimer was tested under the same conditions as the positive control. Figure 2 reports the viability of cells observed at concentrations 0–100 µM of the tested compounds, expressed as a mean percentage of the control (untreated cells, corresponding to G4-PAMAM-NH_2_ and G5K 0 µM) ± SD. Figure 2 shows the polynomial trend line associated with the dose-dependent curve of G5K and the related Equation (6).

At a concentration of 14 µM, the cell viability % of HeLa cells exposed to G4-PAMAM-NH_2_ was lower than 50%; therefore, to obtain the polynomial trend line associated with the dose-dependent curves of G4-PAMAM-NH_2__,_ only the range of concentrations 0–14 µM was considered (Figure 3).

Although more cytotoxic than its analogous lower-generation cationic dendrimer G4K previously tested on the same cell line [32,33], G5K was much less cytotoxic than the commercially available cationic G4-PAMAM-NH_2_ dendrimer, which belongs to the family of PAMAMs, which are the most used dendrimers as drug delivery systems [56]. By using Equations (6) and (7) (Table 2) of the polynomial trend lines associated with the dose-dependent curves of G5K and G4-PAMAM-NH_2_, we calculated the respective LD_50_ values reported in Table 2.

According to the results, G5K was 13.7-fold less cytotoxic than the positive control (Table 2). Moreover, considering the dose of CR232 that was active against SKMEL28 and HeLa cells (10 µM) [22] and considering that to release 10 µM of CR232 in vitro it should be necessary that G5K is only 0.24 µM, we can unequivocally assume that in a possible clinical application of a CR232 formulation based on G5K, G5K NPs will serve only as a reservoir, protector, and solubility enhancer for CR232, devoid of additional and unpleasant toxic effects, since at 0.24 µM of G5K, the percentage of live HeLa cells was 105%.

#### 3.1.5. Preparation of CR232-G5K NPs

We used the highly cationic hydro soluble fifth-generation polyester-based dendrimer G5K containing lysine as reservoir and solubilizing agent to encapsulate CR232. To this end, we performed an association of the widely reported solution casting method and of the solvent diffusion evaporation method [50], which were slightly modified, as previously reported. The non-solubilized CR232 was removed after the encapsulation process by centrifugation, while the residual non-entrapped CR232 was removed by washing. The recovered CR232 was obtained as yellow crystals after recrystallization by Et_2_O/Petroleum ether 1/1, and its identity was confirmed by ATR-FTIR analysis and TLC (results not reported). Interestingly, we obtained water-soluble CR232-loaded G5K NPs without using harmful high boiling and difficult-to-remove organic solvents such as DMSO and co-solvents such as polyethylene glycol 200 (PEG 200), whose safety for humans is questionable [57]. Additionally, no further additives such as stabilizers, surfactants, or emulsifiers (poloxamers), frequently employed in high concentrations to promote the solubilization of insoluble molecules, although they can be toxic to humans [53], were added to the reaction mixture. The encapsulation of CR232 in G5K NPs took place by virtue of the capability of cationic dendrimers with peripheral amine groups (as G5K) to interact by hydrogen bonds and Van der Waals forces with hydrophobic drugs containing nitrogen heteroatoms, such as CR232, thus complexing and solubilizing them in water [58]. Dendrimer G5K was selected as an encapsulating and solubilizing agent due to its hydrolysable and biodegradable inner matrix, which should ensure a low level of systemic toxicity [59]. In addition, in cytotoxic experiments, G5K NPs were proven to be deprived of cytotoxic effects at the antiproliferative concentration reported for CR232 (10 µM) [22], thus assuring that G5K will act only as a drug delivery system and solubility enhancer. On the other hand, the peripheral cationic character of the dendrimer carrier conferred by lysine, in addition to guaranteeing the high hydrophilicity of a drug-loaded G5K-based formulation, would promote its interaction with the tumour cell surface more negatively than that of normal cells and like bacteria [60], which are susceptible to cationic macromolecules [61,62]. Consequently, the presence of the cationic shell provided by G5K would favour the selective cytotoxic activity of CR232 towards cancer cells. Finally, following the electrostatic interactions of the G5K cationic envelope with cancer cell membranes, the cell up-take of the CR232-loaded G5K NPs could occur by endocytosis, which is the most accredited mechanism for the cell internalization of cationic platforms used both for drug delivery and gene therapy [44,45,60]. A meticulous literature research showed that, except for a recent article by us [33], only two articles exist concerning the polymer formulation of pyrazole derivatives [35,36], but only one regarded the improvement of their solubility in water to enhance their activity and make feasible their biomedical application [36]. Different from our dendrimer-based approach that uses dendrimers synthetized by us [33], as in the present study, commercial polymers were used as solubilizing agents, with less appealing results [36]. The present work is the second reported example of successful dendrimer encapsulation of a water-insoluble pyrazole derivative aimed at making it suitable for clinical uses.

#### 3.1.6. NMR Analyses

The presence of CR232 in the structure of CR232-G5K NPs was unequivocally established by ^1^H and ^13^C NMR analyses. The ^1^H NMR spectrum of CR232-G5K NPs also allowed us to calculate the number of moles of CR232 that were loaded per mole of G5K. In Appendix A, Appendix A shows the ^1^H NMR spectrum of CR232-G5K NPs, while Appendix A shows the ^13^C NMR spectrum.

The ^1^H NMR spectrum of CR232-G5K NPs shown in Appendix A, clearly evidenced the presence of signals typical of the structure of CR232 and of signals belonging to G5K. Precisely, a group of signals in the range 7.61–8.18 ppm given by the proton atoms of the phenyl rings of CR232, observed in the spectrum of CR232 (Appendix A), and absent in the spectrum of G5K (Appendix A), was detected very close to the signal exchangeable with D_2_O at 8.20 ppm belonging to the ^ε^NH_3_^+^ groups of G5K. Further, while a very similar signal (exchangeable with D_2_O) belonging to the ^α^NH_3_^+^ groups of G5K, was visible at 8.82 ppm, as in the spectrum of G5K (Appendix A), a very small signal belonging to the aniline NH group of CR232 was detected at 9.92 ppm, as in the spectrum of the pyrazole derivative (Appendix A). Moreover, while in the spectrum of CR232 no peaks were observed under 7.5 ppm, several signals typical of the structure of G5K were observed in the spectrum of CR232-G5K NPs (Appendix A). Specifically, signals at 1.03–1.40 (CH_3_ of the five generations), 1.50–1.99 (CH_2_ of lysine), 2.76 (CH_2_NH_3_^+^ groups of lysine), 3.99 (CHNH_3_^+^ groups of lysine), and 4.10–4.50 ppm (CH_2_O groups of the five generations) were detected. The number of CR232 moles loaded per dendrimer mole was obtained considering the signal belonging to G5K only and given by a known number of proton atoms [CH_2_NH_3_^+^ at 2.76 ppm, accounting for 192 proton atoms per G5K mole (Appendix A)] and a signal belonging to CR232 only and present in a well-separated region of the spectrum at 9.92 ppm (NH of aniline, 1 proton atom per CR232 mole). By dividing the value of the integrals of the signal at 2.76 ppm by 192, the integral value for one proton atom was obtained, and by dividing the value of the integral of the signal at 9.92 ppm by the obtained number, the quantity of CR232 moles loaded per dendrimer mole was obtained (41).

This information was used to determine the molecular formula of CR232-G5K NPs and to calculate their MW, which was in accordance with the MW determined using the results of DL% obtained by UV-Vis analyses (error 0.15%).

The ^13^C NMR spectrum of CR232-G5K NPs further confirmed the success of the encapsulation reactions. In fact, as observed in Appendix A, the spectrum, in addition to the signals typical of the hetero-aromatic nucleus of the 3,4,5-trisubstituted pyrazole of the *p*-disubstituted phenyl rings (119–159 ppm) and of the CN group (84.35 ppm) of CR232 showed all the signals belonging to G5K (higher than 170 and lower than 70 ppm).

#### 3.1.7. UV-Vis Spectra of G5K, CR232, and CR232-G5K

To select the maximum absorption peak of CR232 necessary for constructing the CR232 calibration curve, we acquired its ultraviolet spectrum in the range 230–650 nm in DMSO. Two peaks of absorbance were detected at ʎ_abs_ = 254 and 384 nm, and ʎ_abs_ = 384 nm was preferred for developing the CR232 calibration model and to determine the amount of CR232 in CR232-loaded NPs. Furthermore, to assess the absence of interfering peaks of absorbance of G5K at the chosen ʎ, we also acquired the UV-Vis spectrum of G5K under the same conditions. A single peak of absorbance was detected at ʎ_abs_ = 280 nm, thus confirming the absence of undesired interference. Lastly, the UV-Vis spectrum of CR232-G5K NPs was acquired, detecting a peak of absorbance at ʎ_abs_ = 328 nm that was a value different from that of CR232 and that of G5K, thus establishing that CR232-G5K NPs were stable in DMSO. Appendix A shows the ultraviolet spectra of the three substances.

#### 3.1.8. CR232 Calibration Curve

In Appendix A, Appendix A shows the values of [A] (expressed as [A] mean ± SD) determined for each CR232 concentration injected in the UV-Vis system, the concentrations of CR232 (C_CR232_) used for the UV-Vis analyses, the CR232 concentrations predicted by the calibration model (C_CR232p_), the residuals, and the absolute errors percentages. [A] and C_CR232_ reported in Appendix A were used to develop the CR232 calibration model by the least squares method whose equation was Equation (1). Appendix A shows the obtained linear regression curve.

In addition to consider the value of the coefficient of determination (R^2^ = 0.9964) to confirm the linearity of the developed calibration, its linearity and sensitivity were also established, evaluating the statistical significance of its slope through analysis of variance (ANOVA), performing Fisher’s test. Statistical significance was recognized at a *p*-value < 0.05. Equation (1) was exploited for determining the CR232 concentrations predicted by the model (C_CR232p_) for each sample (Appendix A, third column) that were reported in a dispersion graph vs. C_CR232_ to obtain the regression curve correlating the two sets of data (Appendix A). The existence of a strong correlation between the real and the predicted concentrations of CR232 was evidenced by the high value of R^2^ (0.9908), and by that of the correlation coefficient R (0.9954), thus confirming the goodness of fit of the model.

#### 3.1.9. Determination of CR232 Contained in the CR232-G5K NPs and DL% and EE%

Five aliquots of CR232-G5K at different concentrations were subjected to UV-Vis analysis, obtaining five values of [A] at 328 nm (Table 3), which did not correspond to the ʎ_abs_ = 384 of the untreated CR232 or to the ʎ_abs_ = 280 nm of the empty dendrimer G5K, but corresponded to the ʎ_abs_ of CR232-G5K NPs. Additionally, negligible values of [A] were observed at ʎ_abs_ = 384 nm (CR232), thus establishing that in DMSO, the release of CR232 was insignificant. Therefore, we were forced to use the data of [A] at ʎ_abs_ = 328 nm to determine the related C_CR232_ (µg/mL) concentrations by employing Equation (1) (Table 3). Once data that were considered outliers were removed, the mean concentration ± SD of CR232 was used to obtain the CR232 content in the amount of CR232-G5K weighed for the analysis (6.0 mg) and in the total amount of CR232-G5K obtained from the encapsulation reaction (123.2 mg, Table 3). This latter value allowed us to determine the values of DL%, EE%, and the moles of CR232 loaded for the G5K mole (Table 3). Such values were used to compute the MW of CR232-G5K NPs, which agreed with the value of MW estimated by the ^1^H NMR spectrum of CR232-G5K NPs (Table 3).

The amount ± SD of CR232 in the amount of CR232-G5K analysed (6.0 mg) was 1.9121 ± 0.0389 mg, thus establishing that the total CR232 loaded in the CR232-G5K NPs obtained from the encapsulation reaction resulted in 39.01 ± 0.7987 mg. The DL% was 31.7%, while EE% was 98.3. The DL% of CR232-G5K established that G5K was able to load 41.2 moles of CR232 per dendrimer mole, thus making achievable a great amount of CR232 at the target site of action at a minimal dosage of the CR232 formulation, which should translate into improved activity and reduced systemic toxicity. The DL% observed in this study was slightly higher but in accordance with that observed previously, when we encapsulated, based on a different method, a diverse water-insoluble bioactive pyrazole derivative (BBB4) in a cationic dendrimer of lower generation (G4K) [33]. However, the EE% herein determined was much higher than that observed previously [33]. Several parameters can affect the yield of nanoencapsulation by dendrimers, including their generation, the type of end groups, the surface charge, the core structure, pH, and ambient factors [63]. In this case, the higher generation of G5K, offering the possibility of creating a major number of hydrogen bonds and electrostatic interaction with CR232, could be assumed as the main factor responsible for the higher EE%. Except for our previous study [33], in the only other existing study reporting on the encapsulation of a pyrazole derivative (AMDPC) to enhance its water solubility and activity, by using commercial PEG-PLGA, the authors obtained micelles with DL% = 1.28 and EE% = 64.3 [36], which are values significantly lower than those herein determined for our pyrazole formulation, obtained using a dendrimer synthetized by us.

#### 3.1.10. Determination of CR232-G5K MW

The MW of CR232-G5K NPs was estimated using the number of CR232 moles loaded per dendrimer (G5K) mole obtained by analysing the ^1^H NMR spectrum of CR232-G5K NP. Additionally, the MW of CR232-G5K NPs was estimated using the quantitative results obtained from the UV-Vis analyses. A minimal difference of 0.15% was obtained for the results (Table 3, last column), thus confirming the goodness of the CR232 calibration model and the reliability of the DL% value.

Since the number of CR232 moles entrapped in one mole of G5K estimated by the ^1^H NMR spectrum was 41, the MW of CR232-G5K was determined according to the following Equation (8).
MW_CR232-G5K_ = MW of G5K (30223.9) + 41 × MW of CR232 (339.7) (8)

Since the number of CR232 moles entrapped in one mole of G5K estimated by the UV-Vis analyses was 41.2 ± 0.7, the MW of CR232-G5K was determined according to the following Equation (9).
MW_CR232-G5K_ = MW of G5K (30223.9) + 41.2 ± 0.7 × MW of CR232 (339.7) (9)

If we consider the SD value (0.7), it can be noted that the MW estimated by UV-Vis analyses can vary from 43,981.8 to 44,457.3, a range that exactly includes the MW value (44,153.1) obtained by ^1^H NMR.

#### 3.1.11. CR232-G5K NP Release Profile

The release profile of CR232 from CR232-G5K NPs was studied by a dialysis method in PBS as a receptor medium (pH = 7.4). The CR232 released was determined at fixed points for 24 h firstly by weighing the yellow residues obtained by evaporating the sampled solutions under high vacuum. Secondly, based on analysis using the UV-Vis apparatus previously described, opportunely diluted DMSO solutions obtained by re-suspending the solid residues were obtained. The results were expressed as CR232 cumulative release percentage (CR%) for each time by Equation (10)
(10)$$\mathrm{CR}\;\left(\%\right)=\frac{\mathrm{CR}232\left(t\right)}{\mathrm{CR}232\left(\mathrm{NPs}\right)}\times 100$$
where CR232(*t*) is the amount of CR232 released at (*t*) incubation time, while CR232(NPs) is the total CR232 entrapped in the weight of CR232-G5K NPs analysed, according to the computed DL%.

CR% values are reported in a dispersion graph vs. the incubation times, obtaining the CR232 release profiles (Figure 4).

Although the release profiles obtained by the two measurement ways were very similar, we considered the profile obtained by the UV-Vis analysis, which is one of the most reported techniques to quantify the drugs released from polymeric scaffold, including pyrazole derivatives [36]. As observed in Figure 4, the CR232 release (orange line) had a complex tri-phasic profile, where the usual initial burst release is missing. Notably, three phases of faster release (observed during the first two hours, between the fourth and fifth hour, and in the interval of hours 6–10) were interspersed by two phases of slower release. During the last 14 h, a sustained slow-release phase was observed that led to a practically quantitative release of CR232 after 24 h (99.3%). According to a previous study, the release profile of a pyrazole derivative from PEG/PLGA-based micelles monitored for 48 h [36], different from what was observed by us, was simpler, biphasic, and characterized by an initial fast release in the first few hours, followed by a phase of prolonged sustained release. The maximum release was only 77% after 48 h and 60% after 24 h. The release profile observed for our CR323-loaded formulation can be explained assuming an initial release due to the desorption of non-entrapped CR232 but only adsorbed on the G5K surface. The following two phases of slow release, before two phases of faster release, indicate the necessity for the formulation to interact with the aqueous medium and to hydrate itself, for allowing the further release of CR232.

The final phase of slow, sustained release can be explained by assuming a drug release dependent on the residual CR232 concentration. In this regard, since most of the CR232 (73%) was released in the first 10 h, in the subsequent 14 h, the drug residue was released slowly due to its low residual concentration.

To determine the kinetics of the CR232 release and to investigate the main mechanisms that govern the release of CR232 from CR232-G5K NPs, we used different mathematical models based on different mathematical functions, aimed at describing the drug dissolution profiles. To select the most suitable function and to determine the most suitable mathematical kinetic model describing the dissolution profile of CR232, we firstly fitted the CR% curve data with the zero-order model (which reports in a graph the % cumulative drug release vs. time), first-order model (Ln % cumulative drug remaining vs. time), Hixson–Crowell model (cube root of the % cumulative drug remaining vs. time), Higuchi model (% cumulative drug release vs. square root of time), and Korsmeyer–Peppas model (Ln % cumulative drug release vs. Ln of time) as commonly described [64,65] and as we did in our previous works [32,33,66]. The approximation accuracy of the individual models was assessed in terms of coefficients of determination (R^2^) that have been reported in Appendix A. As the obtained R^2^ values were lower than the acceptable value of 0.95, we considered the additional Weibull model [67,68] that reports in a graph the values of LnLn(100/100-CR%) vs. the Ln of time values. A dispersion graph, whose linear regression showed a significantly higher value of R^2^ (0.9754) was obtained (Appendix A), thus establishing that the CR232 release from the CR232-G5K NPs best fitted the Weibull kinetic.

The Weibull kinetic model can be expressed by the following equation, Equation (11):(11)$$LnLn\frac{C_{0}}{C_{0}-C_{t}}=\beta Lnt+Ln\alpha$$
where *C_t_* is the concentration of drug release in time *t*, *C*_0_ is the initial concentration of the drug present in the nanocomposite system, *t* is the time, *β* is the shape parameter of the dissolution curve, and α is the scale parameter, estimable from the ordinate value (l/*α*), at *t* = 1.

According to the linear regression in Appendix A and Equation (11), the slope of the regression corresponds to the value of *β*, while that of the intercept is the value of *Lnα*. Values of *β* < 0.75 indicate that diffusion mechanisms govern the drug release [68,69], while values in the range 0.75–1.0 indicate a combined mechanism that is frequently encountered in release studies [68,69]. The specific case of *β* = 1 is compatible with the first-order release, in which the drug concentration gradient in the dissolution medium governs the rate of its release. Finally, when *β* > 1, as in the present case, the sigmoid shape of the Weibull function indicates that a complex mechanism governs the release process. The rate of the drug release does not change monotonically, whereas it can initially increase nonlinearly up to an inflection point and thereafter decrease asymptotically and increase again, as observed in the release profile of CR232-G5K NPs developed by us.

#### 3.1.12. Morphology of Particles of G5K and CR232-G5K NPs by SEM

The SEM image of G5K NPs shown in Appendix A indicated a spherical morphology and an average particle size of 200 nm. Concerning CR232-G5K NPs (Appendix A), the SEM image established that the spherical shape did not change following the encapsulation of CR232. A spherical morphology contributes to provide a high surface area, which typically determines retention in a circulation system for longer periods and a slow metabolism, which in turn could translate into improved therapeutic effects [70,71]. Interestingly, the encapsulation of CR232 resulted in a remarkable improvement of the particle size to dimensions of about 500 nm (Appendix A), confirming the DL% results established for a very high amount of the loaded drug.

#### 3.1.13. Potentiometric Titrations

It is generally accepted that a cationic drug carrier (as G5K), to be efficacious in delivering the transported drug inside the cells at the target site escaping phenomena of early inactivation, should have essential requisites [44,45,48,49]. Among others, it should have values of buffer capacity [β = dc(HCl)/dpH] [72] (at physiological pH) and of mean buffer capacity [β mean = dV(HCl)/dpH (1)] [73] (in the pH range 4.5–7.5) sufficiently high to make it capable of escaping from endosome/lysosome compartments, thus shirking lysosomal deactivation. To estimate the buffer capacity of G5K, potentiometric titrations were performed according to Benns et al. [51]. For comparison, potentiometric titrations of CR232-G5K NPs were also performed, and their values of β, as well as β mean were computed. Appendix A) shows the titration curves of G5K and CR232-G5K NPs obtained by reporting in a graph the measured pH values vs. the aliquots of HCl 0.1 N added. Interestingly, for both samples, the titration curves had two end points due to the presence of two different types of primary amine groups in the lysine residues (^α^NH_2_ and ^ε^NH_2_), thus establishing the existence of a two-step protonation process. Subsequently, from titration data of G5K and CR232-G5K NPs, the β values [dc(HCl)/dpH] were computed and are reported in a graph vs. pH values, obtaining curves with a buffer capacity of G5K and of CR232-G5K as a function of pH (Appendix A). Appendix A also shows the bars graph of the β mean values in the pH range 4.5–7.5 for G5K and CR232-G5K compared with those of three different PAMAM dendrimers with and without peripheral amino acids, which are considered standard reference compounds of efficient dendrimer-based drug delivery systems [74]. Both G5K and CR232-G5K NPs had a mean buffer capacity significantly higher than those of the PAMAMs taken as standard reference dendrimers of efficient drug delivery systems. Interestingly, while the curve of the buffer capacity of G5K showed a small maximum in the pH range 6–7, that of CR232-G5K NPs showed a maximum buffer capacity markedly higher. Table 4 shows the values of β (pH range 6–7) and values of β mean (pH range 4.5–7.5) of G5K and CR232-G5K. In this case, the buffer capacity, and the average buffer capacity of G5K and CR232-G5K NPs were compared with data concerning the three PAMAM derivatives previously mentioned and with data concerning commercial branched polyethyleneimine (PEI-*b*).

Although PEI-*b* was the polymer with the higher value of the β mean, both G5K and CR232-G5K NPs displayed values 2.7–11-fold higher than those of the fourth generation PAMAM dendrimers taken as reference dendrimers. Furthermore, the β value of the CR232-dendrimer formulation herein developed was also higher than that of PEI-*b*, thus establishing its high capability to escape lysosomal attack, preventing early inactivation of CR232.

### 3.2. Liposomes

#### 3.2.1. Preparation of CR232-SUVs

We selected lipid NPs (liposomes) as encapsulating and solubilizing agents in our second nanotechnological approach to enhance the physicochemical properties of CR232 and particularly its water-solubility. Briefly, with the use of the film hydration method [52], multilamellar vesicles (MLVs) and then translucent suspensions of small unilamellar vesicles (SUVs), here named CR232-SUVs 5/1, 15/1, and 30/1, were prepared at three different lipids/CR232 ratios as described in Section 2.3.1.

In this case, the use of the PEG derivative DSPE-PEG was necessary to cover the liposome surface and obtain stealth liposomes, with prolonged half-life in blood circulation [75]. In fact, after administration, liposomes are usually recognized by phagocytic cells and are expelled rapidly from the blood. The PEGylation of liposomes can prevent opsonization, thus enhancing their efficiency.

Curiously, after the successful use of a synthetic dendrimer to entrap CR232, completely different lipid-based encapsulating materials, such as liposomes, made of cholesterol, phospholipids, and a PEG derivative, were employed to prepare three different nano-formulations of CR232.

Although they have a limited DL% capacity and solubilizing power respect to cationic dendrimers [76], liposomes have been extensively used in biomedicine, especially to transport and deliver antitumor drugs and antimicrobial agents [40]. Indeed, liposomes can provide several advantages, including the capability to protect the active drugs from environmental factors and early degradation, thus improving the performance features of the transported molecules [41,43]. Additionally, while dendrimer-based drug delivery systems and/or solubilizing agents, even if they perform well, require laborious multi-step synthetic and purification processes, with high amounts of organic solvents, and low-cost and fast production procedures are necessary to prepare drug-loaded liposome formulations. An additional overall merit of liposome-based drugs formulations is a reduced systemic toxicity, [41,43]. Moreover, a study reported that doxorubicin-loaded PEGylated liposomes led to a higher drug concentration inside tumours, with a reduced drug concentration in normal tissues [75]. Liposomes can efficiently encapsulate hydrophobic drugs (as CR232) within their lipid bilayers that are very similar to the structure of cell membranes, thus succeeding in delivering encapsulated drugs simply by fusing with cell membranes [76]. Moreover, it was reported that pharmacokinetic studies in rats revealed that the improvement of oral bioavailability of drugs transported with the liposomes was significantly higher than that obtained with G5-NH_2_ poly-amido-amine (PAMAM) dendrimers [76]. The overall better oral absorption of drug–liposomes as compared to drug–G5-NH_2_ dendrimer complexes arose from the better liposomal solubilization and encapsulation of drugs and from more efficient intracellular drug delivery [76].

Studies concerning the formulations of pyrazole derivatives in liposome vesicles to increase their water solubility and provide them physicochemical properties suitable for therapeutic uses are rare or absent, thus establishing the originality and novelty of our strategy. In fact, only one article was found regarding the synthesis of 1-phenyl pyrazole-3, 5-diamine, 4-[2-(4-methylphenyl) diazenyl] and 1*H*-pyrazole-3 (1), 5-diamine, 4-[2-(4-methylphenyl) diazenyl] and their encapsulation into liposomal chitosan emulsions for textile finishing [35].

#### 3.2.2. Freeze-Drying of Liposome Suspensions (SUVs and CR232-SUVs)

Generally, three phases can be distinguished in the freeze-drying process: freezing, primary drying, and secondary drying. The parameters of each phase can determine the quality of the final product. In the freezing phase, cooling of the sample results in the formation of ice crystals and in the concentration of all solutes and the liposomes [77]. At this stage, the presence of a cryoprotectant forming an amorphous (non-crystalline) matrix in and around the liposomes is essential [77]. Cryoprotectants protect liposomes, preventing their fusion, precluding the rupture of the bilayers by the growth of ice crystals, and preserving the integrity of the bilayers in the absence of water. Sugars can reduce the compressive stress on the adjacent bilayers attracting water (osmotic properties) and providing spacing between them by forming a glassy film in the middle of the neighbouring ones during drying [78,79,80,81]. The preferred cytoprotectants are disaccharides such as sucrose, trehalose, maltose, and lactose. Trehalose, maltose, and lactose have a higher glass transition temperature (Tg) in the dried state (<0.5% residual water) than sucrose, which may be an advantage for the storage stability of the dried product. Furthermore, experiments have proved that in various freeze-drying processes, sucrose and trehalose are the two most used cryoprotectant agents. Accordingly, to detect the best performing and most suitable cryoprotectant for the liposomes prepared by us, we investigated the efficiency both of trehalose and sucrose in preventing liposome collapse. Notably, the sugars were added to the liposome suspensions in equal concentration. As an example, Appendix A shows the solid liposomes obtained by freeze-drying the CR232-SUVs suspension 30/1, but similar results were obtained also freeze-drying the CR232-SUVs suspensions 15/1, 5/1 and the suspension of the empty liposomes. As expected, the glass container 6, containing the liposome powder obtained without cryoprotectant, showed a total collapsed cake. On the contrary, the glass container 7 that contained the liposome powder obtained by adding trehalose, showed no evidence of collapse, while the glass container 9 containing the power obtained using sucrose showed a partially collapsed cake, thus establishing that trehalose was the more suitable cryoprotectant for our liposomes. Indeed, compared with sucrose, trehalose has a higher glass transition temperature (−29 °C), thus being less likely to form ice crystals, which is detrimental for the liposome bilayers. Further, trehalose has a magical hydration capacity, so that the number of non-frozen water molecules around trehalose per glucose unit is the largest number among sugars. Trehalose cryoprotectant can form a more rigid trehalose/water structure and has stronger anti-freeze ability, thus being better than sucrose as a cryoprotectant. Studies have shown that by using glucose, the particle size of liposomes changes after freeze-drying, becoming the largest, and the protection effect is the worst; the lyophilized liposomes with trehalose as a protective agent have the smallest particle size change and the best protection effect. Note that, as reported, a 10% concentration of the trehalose cryoprotectant, as used in this study, provided the best protection for lyophilized liposomes [82].

#### 3.2.3. Determination of the CR232 Concentration in the Prepared CR232-SUVs and the EE% and DL%

The concentration of CR232 in the CR232-SUV suspensions with a lipid/CR232 ratio of 5/1, 15/1, and 30/1 was determined on the CR232-SUV suspensions (3 mL). The CR232 concentration, which corresponded to the water solubility of CR232 present in the vescicles, was quantified by UV-Vis analysis, detecting [A] values at ʎ_abs_ = 384 nm. Then, the EE% for each formulation was determined using Equation (4) (Table 5).

As expected, the EE% increased with the increase in the lipids/CR232 ratio, reaching a very high value (90%) for CR232-SUV 30/1. The EE% value was always lower than that obtained using dendrimer G5K as a solubilizing agent. Unfortunately, comparisons between the EE% values obtained by us with previously reported data were impossible because the unique study already published regarding the liposomal encapsulation of a pyrazole derivative [35] did not report EE% investigations, thus establishing the originality of our approach and of our characterization. Following the lyophilization of CR232-SUVs, the DL% was determined according to Equation (5) (Table 4). The DL% values of the liposome-based formulations of CR232 were low compared with the DL% determined for CR232-G5K NPs. As they were within the range 3.95–4.34, the DL% values of the liposomes were 7.3–8.0 lower than that obtained using G5K NPs as encapsulating agents. The DL% capacity obtained by us using lipid-based biocompatible NPs as liposomes was higher than that obtained previously by Sun et al., who encapsulated AMDPC in PEG-PLGA NPs [36]. In that study, micelles with DL% = 1.28 were obtained, i.e., 3.1–3.4-fold lower than our results.

#### 3.2.4. CR232-SUV Release Profile

The release profile of CR232 from the CR232-SUV formulation 30/1 was selected because it resulted in the highest DL%, and EE% was studied by dialysis in PBS as a receptor medium (pH = 7.4). The results were expressed as the CR232 cumulative release percentage (CR%) ± SD according to Equation (10), and CR% values were reported in a dispersion graph vs. the incubation times to obtain the release profile of CR232 (Figure 5).

The literature reports that drug release profiles from liposomes characteristically show an initial fast drug loss (burst release) followed by slower rates of drug loss [83,84]. It is assumed that, while the initial burst release is usually related to the drug detachment from the liposomal surface, the subsequent slow release is due to a sustained drug release from the inner lamellae of liposomes. On the contrary, as observed in Figure 5, the in vitro release study of CR232 from the liposome-based formulation showed no burst effect, indicating that CR232 was mainly located within the bilayer lipid structure of the liposomes, also stabilized by cholesterol, and the drug transport out of the liposomes was driven mainly by a diffusion-controlled mechanism independent from the drug concentration. The further slight reduction of the release rate between 6–16 h could be the effect of the agglomeration of more PEG over the liposome surface after a particular time interval that further stabilized the liposomes. Collectively, the CR232 release was slow according to the slower release reported for PEGylated liposomes due to the fast hydration process occurring due to the presence of PEG on the surface of the particles. The above results suggest that the drug would be stable in blood circulation and would be released slowly at the target site, thus indicating that our PEGylated liposomal formulation meets the requirements for an effective drug delivery system.

As for CR232-G5K NPs, to investigate the kinetics and the main mechanisms that govern the release of CR232 from CR232-SUVs and according to what is suggested in the literature [67], we fitted the data of the CR% curve with some mathematical models including zero-order, first-order, Higuchi, Korsmeyer–Peppas, and the Weibull model. In this case, according to the R^2^ values (Appendix A) the CR232 release from liposomes best fitted the zero-order kinetic model (Appendix A).

The zero-order kinetic model can be expressed by the following equation, Equation (12):(12)$$D_{t}=D_{0}+K_{0}t$$
where *D_t_* is the amount of drug dissolved in time *t*, *D*_0_ is the initial amount of drug in the solution, and *K*_0_ is the zero-order release constant.

As observed in Appendix A, the zero-order kinetic dispersion graph was obtained by plotting the CR% vs. times that would yield a trend line with a slope corresponding to the zero-order release constant (1.3747) and an intercept corresponding to the initial amount of drug in solution (2.1414). The zero-order kinetics model is typical of formulations that do not disaggregate, release drugs slowly, and transport drugs poorly soluble in water [64] (as in this case). The drug release rate is constant over time and independent of the drug concentration, thus confirming our empirical assumption derived from the simple observation of the CR232 release profile in Figure 5 and thus establishing that liposomes can act as reservoir systems for continuous drug delivery.

### 3.3. Dendrimer NPs and Liposomes

#### 3.3.1. ATR-FTIR Spectroscopy

The success of both encapsulation strategies was firstly assessed qualitatively, acquiring the ATR-FTIR spectra of pristine CR232, empty dendrimer G5K, empty liposomes (SUVs), purified CR232-G5K NPs, and purified freeze-dried CR232-SUVs. The obtained spectra were firstly compared by simple observation.

Figure 6 shows the spectra of G5K (green line), CR232 (red line), and CR232-G5K (light blue line). As expected, the spectrum of CR232-G5K NPs was very similar to that of the encapsulating agent (G5K) due to the well-exposed functional groups of the dendrimer external envelope, which provided high absorbance and intense bands overlapping most of the bands given by the functional groups of CR232 packed inside.

Based on observations of the regions of the spectrum inside the rectangles (Figure 6a), typical bands of CR232, such as those at 2224 cm^−1^ (stretching CN), 1600 cm^−1^ (aromatic rings), and at 3224 cm^−1^ (stretching NH), not observed in the spectrum of G5K, were well detected in the spectrum of CR232-G5K at 2226, 1602, and 3331 cm^−1^, thus confirming the presence of CR232 in the prepared nanocomposite formulation. Additionally, the typical intense band due to the stretching C=O (1744 cm^−1^) of the ester groups of G5K, not observed in the spectrum of CR232, was well visible in that of CR232-G5K NPs at 1735 cm^−1^. Figure 6b shows the magnification of the spectral region in the range 2250–1450 cm^−1^, better showing the CN bands typical of the pyrazole derivative, the ester bands typical of the structure of G5K, and other smaller bands typical of CR232 visible both in the spectrum of CR232 (1600 cm^−1^) and in that of CR232-G5K NPs (1602 cm^−1^).

Figure 7 shows the spectra of SUVs (black line), CR232 (red line), CR232-SUVs 5/1 (light blue line), CR232-SUVs 15/1 (fuchsia line), and CR232-SUVs 30/1 (green line). As expected, the spectra of all CR232-SUVs were very similar to those of SUVs due to the low content of CR232 and to the better exposition of functional groups of the lipidic external envelope that gave very intense bands overlapping most of the bands given by the functional groups of CR232 hidden inside.

Through careful observation of the spectra of the liposome-based formulations that resulted in the high EE% and DL% (CR232-SUVs 15/1 and 30/1), even if very small, typical bands of CR232 at 2225–2226 cm^−1^ (stretching CN) and at 1604 cm^−1^ (CH=CH stretching phenyl rings) were detected. Figure 7b shows the magnifications of the most significant region of the spectrum showed in Figure 7a. To observe the spectra of SUVs and of CR232-SUVs in separate images, copies of the ATR-FTIR spectra of SUVs (Appendix A) and of CR232-SUVs 5/1, 15/1, and 30/1 (Appendix A) are available in Appendix A.

#### 3.3.2. PCA of the ATR-FTIR Spectral Data

The presence of CR232 in the prepared CR232-loaded NPs was unequivocally assessed applying multivariate analysis (MVA) to the FTIR spectral data, employing PCA, which is a chemometric tool that transforms data sets made of thousands of variables into a reduced number of new variables called principal components (PCs) namely, PC1, PC2, PC3, etc., based on the decreasing percentage of variance explained [32,33,85,86]. When the spectral data of a series of chemical samples are processed by PCA, a score plot is obtained, which gives information concerning their physicochemical composition, evidencing chemically and/or structurally similar and dissimilar compounds, based on their reciprocal position in the plot. In the present case, different score plots showing the reciprocal positions of CR232, G5K, CR232-G5K NPs, SUVs, and CR232-SUVs were obtained. We have reported the score plots that provided the most significant information. While Figure 8 shows the score plot of PC1 vs. PC2 concerning CR232, G5K, and CR232-G5K NPs, Figure 9 shows the score plot of PC1 vs. PC2 concerning CR232, SUVs, and CR232-SUVs.

Accordingly, CR232-G5K NPs were located with a low positive score on PC1 and a high positive score on PC2. Importantly, CR232-G5K NPs were positioned closer to CR232 (score around 0 on PC1 and close to 10 on PC2) than to G5K, which had a high positive score on PC1 and a negative score on PC2, thus evidencing that the chemical composition of CR232-G5K NPs was strongly affected by that of the encapsulated drug (CR232). These findings proved the presence of a very high amount of CR232 in the obtained CR232-G5K NPs, as was confirmed by the ^1^H NMR analysis and UV-Vis determinations. By calculating the ratio percentage between the scores (in centimeters) on PC2 of CR232 (1.95 cm) and of CR232-G5K NPs (6.15 cm), we predicted the DL% of CR232-G5K NPs, which, fascinatingly, was identical to that obtained by UV-Vis analysis (31.7% vs. 31.7 ± 0.6%).

Concerning the CR232-SUVs, they had positive scores on PC1 and negative scores on PC2 very close each other, very close to the location of the empty liposomes, and very distant from CR232, which had positive scores on PC1 and PC2, thus evidencing a chemical composition where the CR232 contribution was very low, resulting in DL% values that were significantly lower than those of CR232-G5K NPs. Additionally, the propinquity of the three CR232-SUVs suggested very similar DL% values, as was confirmed by the DL% values determined by UV-Vis analyses. However, the reciprocal positions of CR232-liposomes 30/1, 15/1, and 5/1 closer to CR232 than the empty liposomes (both on PC1 and PC2) in such a specific order evidenced an albeit minimal difference among their structures, as well as their higher similarity with CR232 than SUVs, and agreed with the data of EE% and DL%, which decreased from formulation 30/1 (the closest to CR232) to 5/1 (the furthest from CR232).

The most significant score plot (PC1 vs. PC3) obtained from the PCA the spectral data of all samples together (CR232, G5K, CR232-G5K NPs, SUVs, and CR232-SUVs 30/1, 15/1, and 5/1) was observed in Appendix A. Interestingly, the different typologies in terms of chemical composition were well separated both on PC1 and PC3. Concerning this, all liposome-based materials were clustered at positive low scores very distant from all other samples. The very high distance from G5K on PC1 evidenced the strong difference between dendrimer-based and liposome-based NPs. Different from CR232 and CR232-G5K NPs that were located at negative scores on PC3, both G5K and liposomes were in the right sector of the plot at positive scores on PC3, thus evidencing the empty status of G5K (which did not contain CR232) and the low content of CR232 in the liposome formulations. On the contrary, the location of CR232-G5K NPs in the left sector of the plot such as CR232 evidenced the high contribution of CR232 in the chemical composition of CR232-G5K NPs, as confirmed by the high number of moles of CR232 loaded for the dendrimer mole.

#### 3.3.3. Water Solubility Determinations

The water solubility values of CR232, CR232-G5K, and the nanoengineered CR232 contained in CR232-G5K NPs according to the DL% value obtained by UV-Vis determinations and of the freeze-dried CR232-SUVs that resulted in the highest EE% and DL% (lipids/CR232 30/1) were obtained by the shake-flask method [33,50,53]. However, the water solubility of CR232 contained in all prepared liposome-based formulations was measured directly on the aqueous suspensions obtained by hydrating the lipid films of the CR232-SUVs. Table 6 shows the results.

In a study by Plöger et al., based on their solubility at three different pH values, 16 pharmacologically active substances were classified as highly soluble and not highly soluble molecules. According to the data for the pH value of 6.8, drugs with solubility in the range 1.73–5.13 mg/mL were classified as highly soluble [87]. Following this classification, CR232-G5K NPs can be classified as highly soluble, while the water solubility of CR232 contained in the formulation was slightly inferior to the minimum limit to be classified highly soluble. Considering that the water solubility of the untreated CR232 was 2.25 µg/mL, the water solubility of CR232-G5K NPs and of CR232 contained in G5K NPs was 2311 and 733-fold higher, respectively, than that of pristine CR232. The water solubility of the CR232-loaded dendrimer NPs developed in this study, as well as that of the pyrazole-derivative contained in the NPs, was slightly lower than that obtained in our previous study [33]. Since the pristine pyrazole BBB4 of that study was 26.7-fold more soluble than CR232, the water-solubility improvement achieved in this study was markedly higher than that achieved previously, probably due to the higher number of peripheral cationic groups of G5K. Furthermore, by using a dendrimer as an encapsulating and solubilizing agent in place of traditional polymers as reported by Sun and colleagues [36], we obtained pyrazole-loaded NPs with a water solubility that was 104-fold higher than that of their micelles. Additionally, due to the higher value of DL% determined for CR232-G5K NPs (31.7% vs. 1.28% of micelles), the water solubility of the CR232 pyrazole derivative contained in our formulation was 2578-fold higher than that of the pyrazole derivative contained in the micelles developed by Sun [36]. Liposomes were significantly less efficient than G5K NPs in enhancing the water-solubility of CR232. The water-solubility of CR232 contained in liposomes was in the range 0.07–0.10 mg/mL, thus allowing an improvement markedly lower than that obtained using G5K (16.5–23.5-fold lower). Different from CR232-G5K NPs obtained using G5K of synthetic origin, CR232-SUVs were derived by mixing lipids of natural origin and normally present in the membrane of human cells, thus assuring a high level of biocompatibility and a high possibility to enter cells through micropinocytosis or passive diffusion with or without the occurrence of fusion. Note that liposomes may directly interact with the cell or exchange lipid fragments with the cell membrane through protein-mediated processes [88]. Additionally, always taking the work of Sun et al. as reference study [36], the water solubility of the CR232 contained in the liposome suspensions was 109.4–156.3-fold higher than that previously reported [36]. Interestingly, as for the dendrimer-based CR232 formulation, for the liposome-based ones, the water solubility of the complex was markedly higher than that of CR232 contained in the solubilized NPs. If we consider the water solubility of the freeze-dried liposome-based formulation with the highest DL% and EE% (lipids/CR232 30/1), in addition to being only 1.3-fold lower than that of CR232-G5K NPS, it was 1764.4-fold higher than that of pristine CR232, whereas if we consider the water solubility of CR232 contained in the same liposome-based formulation, in addition to being 20.6-fold lower than that of CR232 contained in CR232-G5K NPs, it was only 35.6-fold higher than that of CR232.

#### 3.3.4. Dynamic Light-Scattering Analysis (DLS)

Table 7 shows the results obtained from DLS analyses of G5K NPs, CR232-G5K NPs, and CR232-SUVs concerning their size (Z-ave, nm), polydispersity index (PDI), and Zeta potential (ζ-p).

Appendix A shows the representative particles size and ζ-p distributions selected among the acquired analyses of the empty dendrimer (G5K) (Appendix A), of the CR232-loaded dendrimer (CR232-G5K) (Appendix A), and of the CR232-loaded liposomes (CR232-SUVs) (Appendix A).

Concerning G5K NPs, the mean particle size was 175.7, and the mean PDI was 0.129, whereas for CR232-G5K NPs, the corresponding values were 529.7 and 0.472, thus establishing a marked increase in both size and PDI, following the encapsulation of CR232 and confirming the results of SEM analyses. Concerning particle size, a similar phenomenon was already observed by us when we encapsulated ursolic acid (UA) in the fourth generation cationic dendrimer G4K [32]. In that study, as in the present one, the encapsulation reaction led to UA-G4K NPs with high DL% (32.4), and a high number of moles of UA loaded per dendrimer mole (33) [32], which resulted in a considerable increase in particle size. For biomedical applications, a delivery system with a particle size around 100 nm is advised [89], and drug-loaded gelatin NPs with a size even higher than that of CR232-G5K NPs (753.3 nm) had high bioavailability [90]. The major concern about CR232-G5K NPs may be their high PDI, which may indicate instability in water solutions and a tendency to form aggregates. Nevertheless, the high value of ζ-p (>30 mV), which is considered a reference value to predict the possible behavior of NPs in water solution, opposes the previous hypothesis. Indeed, ζ-p values around ±30 mV, as observed for CR232-G5K NPs, usually assure good physical stability of the formulation in water solutions with no tendency to form aggregates [91]. Additionally, based on the studies published so far, the internalization of large cationic NPs, such as those developed in this study, is more efficient than that of neutral and anionic NPs [92,93,94,95,96]. Notably, it was found that after electrostatic interaction with anionic components of cells membrane as phospholipids, which with tumor cells are even stronger than with normal cells, thus helping a selective anticancer action, cationic NPs can be internalized by pore formation, micropinocytosis, as well as clathrin- and dynamin-dependent endocytosis [97].

As expected, the size and PDI values of the liposome-based CR232-loaded formulations were very low and like those of the empty dendrimer due to the extrusion process, which is a procedure forcing the dispersed phase containing large liposomes to pass through a membrane or a filter with a uniform pore size distribution, thus generating a homogeneous population of smaller vesicles [98]. Indeed, since CR232-SUVs had a PDI value below 0.2 (0.118), the size of CR232-SUVs NPs can be undoubtedly considered monodispersed. Additionally, although lower than those of CR232-G5K NPs, the ζ-p values were positive (+17.8 mV). According to what has been reported, a major problem in the use of liposomes for the delivery of drugs by injection into the blood stream is the specific uptake of the liposomes by the reticuloendothelial system (RES) [99]. In this regard, the slight positive value of the ζ-p and the steric hindrance of the PEG chains should prevent the uptake of CR232-SUVs by RES, thus assuring a high circulation time [100].

### 3.4. Summary of the Main Physicochemical Properties of CR232-G5K NPs and CR232-SUVs for Easy Comparison

Table 8 shows the results and data obtained by the experiments performed for a complete physicochemical characterization of the CR232-loaded dendrimer and lipid-based NPs.

## 4. Conclusions and Future Perspectives

In this study, with the future perspective of developing a new water-soluble pyrazole-based therapeutic in vivo administrable without using harmful fallbacks, the antiproliferative pyrazole derivative CR232 was nanotechnologically modified by performing two different solubilizing strategies. The first one involved the use of a synthetic high-generation cationic dendrimer, while the second one concerned the exploitation of biocompatible natural lipids as encapsulating and solubilizing agents. CR232 was firstly physically entrapped into a biodegradable and non-cytotoxic cationic dendrimer of the fifth generation (G5K) synthetized by us, thus obtaining water-soluble dendrimer NPs loaded with CR232 (namely, CR232-G54K NPs). Secondly, CR232 was encapsulated in liposomes made of phospholipids, cholesterol, and a PEG derivative, obtaining three different CR232-loaded liposomes suspensions (namely, CR232-SUVs), which differed in the nominal lipids/CR232 ratio (5/1, 15/1, and 30/1).

According to the typology of the formulation, several typical analyses were performed to characterize both CR232-G5K NPs and CR232-SUVs, to determine their chemical compositions, and to confirm their structure. ATR-FTIR analyses confirmed the success of both encapsulation reactions. Additionally, NMR analyses further established the success of the encapsulation reaction performed to obtain CR232-G5K NPs and helped us to determine the number of moles of CR232 that were loaded per mole of G5K. These data were used to compute the MW of CR232-G5K NPs. Additionally, the MW of CR232-G5K was computed considering the DL% value determined by UV-Vis analyses, obtaining a value that perfectly fitted that obtained by the ^1^H NMR spectrum (error 0.15%). The DL% of CR232-SUVs was determined using freeze-dried CR232-SUVs by UV-Vis analyses. Interestingly, the DL% of our formulations were 24.8-fold (CR232-G5K NPs) and 3.1–3.4-fold (CR232-SUVs) higher than those of the micellar NPs obtained encapsulating a pyrazole derivative in commercial copolymers (PEG-PLGA) recently reported. The same micelles also resulted in an EE% value lower than those of both our formulations, by 1.5-times (CR232-G5K NPs) and 1.4-times (CR232-SUVs 30/1). While the release of CR232 from CR232-G5K was insignificant in DMSO, as confirmed by UV-Vis experiments, CR232-G5K NPs showed a triphasic release profile governed by Weibull kinetics and by complex mechanisms variable in time, and a quantitative release after 24 h, under physiological conditions. Such quantitative release, in future biological evaluations, will assure a high concentration of CR232 at the target site at a low dosage of CR232-G5K NPs. However, the CR232-SUVs release profile was slow and was ruled by zero-order kinetics, thus assuring a constant and sustained release of CR232, independent of the CR232 concentration. From DLS experiments, it was established that both CR232-G5K and CR232-SUVs particles were nanosized, with lower dimensions and PDI for CR232-SUVs, positive ζ-p for both formulations, with higher values for CR232-G5K NPs, thus assuring a low systemic toxicity for SUVs and a low tendency to form aggregates, for G5K NPs. Micrographs of the dendrimer formulation obtained by SEM showed a spherical morphology with a high surface area, which typically translates into a high systemic residence time and bio-efficiency. The positive surface of both formulations in future biological evaluations of cancer cells will promote the electrostatic interactions with the surface of cancer cells that are known to be more negative than those of normal cells, thus assuring higher selectivity for malignant cells and low cytotoxicity towards healthy ones. Titration experiments performed on CR232-G5K NPs to determine the buffer capacity, an essential parameter to escape lysosomal attack and to avoid early inactivation, demonstrated a buffer capacity value higher than that of PEI-*b*, which is considered a standard reference polymer of well-functioning and efficient drug delivery systems.

Collectively, G5K NPs were more efficient than liposomes in improving the water-solubility of CR232 and provided NPs with DL% and EE% values markedly higher. Both typologies of CR232 formulations herein developed demonstrated values of DL%, EE%, and water-solubility much higher than those reported for previously prepared pyrazole-based micelles. Importantly, both types of CR232 formulations were obtained without the use of harmful high-boiling and difficult to completely remove solvents such as DMSO and without using surfactants, and/or emulsifiers, which are dangerous for humans. Based on the nanotechnological manipulation of CR232 using dendrimers and liposomes, the water-insoluble CR232, devoid of scientific relevance because it is not clinically applicable, was converted into formulations with a respective water-solubility 2311- and 1764-fold higher than that of untreated CR232 and is worthy of further biological investigations. Once successful in having obtained water-soluble forms of CR232, investigations to re-evaluate the antiproliferative effects of CR232-G5K NPs and of CR232-SUVs 30/1 and to assess the influence of the nanomaterial-based reservoir on the bioactivity of CR232 will be the subject of our next work.

## Data Availability

All data concerning this study are contained in the present manuscript or in previous articles whose references have been provided.

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
