# Peer review of "Successful Dendrimer and Liposome-Based Strategies to Solubilize an Antiproliferative Pyrazole Otherwise Not Clinically Applicable"

_nanomaterials, 2022, doi:10.3390/nano12020233_

Round 1

Reviewer 1 Report

The paper presented by Alfei et al present an extensive study of the incorporation of an hydrophobic drug into dendrimeric NPs or liposomes.

The paper seem's to be a PhD dissertation and not a journal paper. The authors should cut off some of the results to make the paper clearer. The mixture of the results between dendrimers and liposomes is confusing.

The paper should be, in my opinion, fully rewriten to simplify the message which is of interest. A paper with just the FT-IR analysis as a core could be enough.

For exemple the SEM images a low magnification is not usefull at all (also the quality is too poor)

Author Response

The paper presented by Alfei et al present an extensive study of the incorporation of an hydrophobic drug into dendrimeric NPs or liposomes.

The paper seem's to be a PhD dissertation and not a journal paper. The authors should cut off some of the results to make the paper clearer. The mixture of the results between dendrimers and liposomes is confusing.

The paper should be, in my opinion, fully rewriten to simplify the message which is of interest. A paper with just the FT-IR analysis as a core could be enough.

For exemple the SEM images a low magnification is not usefull at all (also the quality is too poor)

We are very thankful to the Reviewer for his suggestions which have been fully applied.

Indeed, the manuscript has been completely re-written with particular attention to the requests of the Reviewer.

Notably, some of the results have been permanently removed by the main text or moved to the Supplementary Materials. Also, Figures, Schemes and Tables have been reduced by seven, one and three items, respectively. Making this, SEM images have been removed. The manuscript resulted shortened by more than six pages.

As suggested, the experimental procedures and analyses concerning dendrimer NPs were divided from those concerning liposomes creating individual Sections. The same has been applied also to the results and to the discussion.

In doing this, we had to reorder most of the references both in the main text and in the list of references. As this extensive review of the manuscript has completely changed its form, the manuscript with the revisions highlighted was illegible and highly confusing. Consequently, regarding the requests of this Reviewer, we have preferred to provide the final form, where only the changes requested by the other Reviewers appear highlighted. Anyway, if the Reviewer still wants to see the manuscript with the changes made highlighted, we will be ready to provide it to him.

Additionally, since in the Reviewer’s report, we noticed that an extensive editing of English language and style was required, the manuscript has been revised by the native English teacher, Prof. Deidre Kants, who works for both the University of Genoa and of Pavia.

Reviewer 2 Report

This manuscript reported solubilizing strategies for enhancing solubility of an antiproliferative pyrazole based on dendrimer and liposomes. This work is not novel, and not appealing either. This manuscript is not well organized, and the figures/tables are not displayed in a scientific way, such as Figure 1, 2, 6, 7, 14 and Table 10. The whole manuscript is tedious with few important data highlighted. Authors used G5k as a carrier for solubilizing pyrazole, while the safety of G5k is concerning. And it is very confusing to use paclitaxel as controls because paclitaxel is commonly regarded as a highly cytotoxic drug. Slightly lower cytotoxicity than paclitaxel does not mean G5k is a safe drug carrier. In a word, this manuscript in this version is too premature to be published in this journal.

Author Response

This manuscript reported solubilizing strategies for enhancing solubility of an antiproliferative pyrazole based on dendrimer and liposomes. This work is not novel, and not appealing either. This manuscript is not well organized, and the figures/tables are not displayed in a scientific way, such as Figure 1, 2, 6, 7, 14 and Table 10. The whole manuscript is tedious with few important data highlighted. Authors used G5k as a carrier for solubilizing pyrazole, while the safety of G5K is concerning. And it is very confusing to use paclitaxel as controls because paclitaxel is commonly regarded as a highly cytotoxic drug. Slightly lower cytotoxicity than paclitaxel does not mean G5k is a safe drug carrier. In a word, this manuscript in this version is too premature to be published in this journal.

We are grateful to the Reviewer for having considered our study, which, unfortunately did not meet its approval criteria. In this regard, we are very sorry about this, but we have strong arguments to counter each of the critical points (according to the Reviewer) that have been notified to us, except for the Reviewer comment concerning the use of Paclitaxel as control. In fact, concerning this, we agree with the Reviewer, and we thank him for having encouraged us to reconsider our choice.

As suggested by the Reviewer, we thought that structural analogues of G5K widely used as drug delivery systems could be more suitable substances as positive control. Particularly, we have chosen a commercially available poly-amidoamine dendrimer (G4-PAMAM-NH2) belonging to the well-known family of dendrimers (PAMAMs) which, like G5K, are cationic due to the presence of peripheral amino groups. Notably, G4-PAMAM-NH2 was tested on HeLa cells under the same conditions as G5K, the dose-dependent curve of G4-PAMAM-NH2 was obtained, and the related polynomial tendency line was used to compute the LD50 of G4-PAMAM-NH2. Accordingly, both the experimental Section 2.2.4. (Lines 259-261) and the results and discussion Section 3.1.4. (Lines 599-602, 603-612 and Table 2) have been modified, as well as Figure 2a and 2b. We further thank the Reviewer for his suggestion which provided us with results that were even more favourable for G5K than those previously obtained considering Paclitaxel. Concerning other comments, following our rebuttals.

The Reviewer defined our manuscript “not novel, and not appealing either”.

Regarding this, we are confident to rebut the above-mentioned and unfounded comment by reporting and commenting some sentences present in the Introduction of the original version of our manuscript.

Note that, the novelty and relevance of our study and of the obtained results is underlined also in several other parts of the Results and Discussion Section.

From Introduction:

“To our knowledge…. only two studies exist in literature concerning the encapsulation of bioactive pyrazole derivatives in nanoparticles (NPs) [35,36], and only one regards the application of nanotechnologies to enhance the pyrazole water solubility [36]. Particularly, Sun et al. [36] have recently encapsulated the pyrazole derivative 6-amino-4-(2-hydroxyphenyl)-3-methyl-1,4-dihydropyrano [2,3-c] pyrazole-5-carbonitrile (AMDPC), found active as anticancer agent, in the poly (ethylene glycol) methyl ether-block-poly(lactide-co-glycolide) (PEG-PLGA), obtaining micelles which gave clear water solutions at 0.05 mg/mL, while at the same concentration pristine AMDPC was insoluble. Despite the authors succeed in improving the AMDPC water solubility, considering the very low DL% (DL = 1.28%) obtained in this study, the water solubility achieved for the encapsulated AMDPC was almost insignificant (6.4 × 10-4 mg/mL), thus making not feasible or difficult the in vivo administration of the obtained AMDPC formulation, without resorting to harmful solvents, as DMSO.

As reported in the above sentences, the encapsulation of bioactive pyrazole derivatives in nanomaterials to improve their water-solubility is per se a novelty, and to use non-commercial dendrimers (as in our study) is even more so. Additionally, although the attempt of Sun et al. managed to solubilize AMDPC by its encapsulation in PEG-PLGA, the obtained DL % and water-solubility were very low. Differently, with our completely innovative for pyrazoles dendrimer-based strategy, the DL% was of 32% and the water-solubility of 5.2 mg/mL. Moreover, the overall merit of the present study consists of having used a synthetic dendrimer designed, prepared, and characterized by us, instead of easy-to-obtain commercial polymers.

Moreover, always taken from the Introduction:

In addition, since in literature we have found no study regarding the use of liposomes to enhance the water-solubility of bioactive pyrazole derivatives, we though it could be interesting to explore also this nanotechnological approach to improve the solubility profile of CR232. Specifically, the only one study we found regarded the encapsulation of 1-phenyl pyrazole-3, 5-diamine, 4-[2-(4-methylphenyl)diazenyl] and 1H- pyrazole-3 (1), 5-diamine, 4-[2-(4-methylphenyl)diazenyl] into liposomal chitosan emulsions for textile finishing [35].

As evidenced in the above sentences, the encapsulation of bioactive pyrazole derivatives in liposomes to improve their water-solubility is totally unreported, thus confirming the originality and novelty of this approach.

Additionally, we can state that the two nanotechnological approaches developed in our work to improve the physicochemical characteristics of pyrazole CR232, as well as being innovative, have been successful, since the prepared nano-formulations have both proved to be considerably performing.

What else does the Reviewer need to certify the novelty and the scientific relevance of our study?

This manuscript is not well organized, and the figures/tables are not displayed in a scientific way, such as Figure 1, 2, 6, 7, 14 and Table 10.

Concerning the organization of the work, also according to another Reviewer, which, differently from this one, kindly provided his suggestions for a better organization, the manuscript has been completely re-organized.

Concerning Figures 1, 2, 6, 7, 14 and Table 10, we do not understand what the Reviewer means by saying that “are not displayed in a scientific way”.

Note that, Figure 1 is a chemical structure obtained using ChemDraw Ultra 7.0 software, a well-known software, worldwide used to draw the chemical structure of molecules. So, what's unscientific about the image in Figure 1?

Original Figure 2, but also Figure 2 now present in the revised version of our manuscript, show a graph obtained using Microsoft excel software, in which the cells viability (%) of HeLa cells exposed for 24 h to G5K and to a control molecule at concentration 0–100 µM has been reported as a function of drugs concentration. The related polynomial tendency lines with the equations which were necessary to determine the LD50 of the considered samples were also described in Figure 2. So, what's unscientific about the image in the original Figure 2?

Figure 6 and Figure 7 of the original manuscript (Figure 8 and Figure 9 in the revised version of the manuscript) show the results, as score plots, obtained performing the principal components analyses (PCA) on the spectral data obtained analysing both dendrimer and liposome materials by ATR-FTIR spectroscopy. The score plots were provided by PAST statistical software (paleontological statistics software package for education and data analysis, free down-loadable online, at: https://past.en.lo4d.com/windows), a well-known software extensively used by scientists to statistically process data. So, what's unscientific about the images in Figure 6 and 7? Perhaps the Reviewer does not know the scientific relevance of PCA?

Figure 14 shows the mean particle size (Z ave) distributions and zeta potentials (ζ-p) distributions obtained performing the DLS analyses on all nanoparticles reported in the present manuscript, by using a Malvern Nano ZS90 light scattering apparatus (Malvern Instruments Ltd., Worcestershire, UK), which provided the images. So, what's unscientific about the images in Figure 14?

Finally, Table 10 summarizes the results reported in the manuscript providing the readers with a useful tool to make proper comparisons between the two different formulations developed. Note that a similar Table was inserted in our previous work recently published on Biomedicines, without eliciting any negative comment from the Reviewers, who deemed it scientifically valid.

The whole manuscript is tedious with few important data highlighted.

While rebutting this unreal comment by the Reviewer, we suggest him to better consider just Table 10 which collects the several interesting results reported in our work deriving from several physicochemical analyses, some of which rarely or never performed for characterizing NPs and drug-loaded NPs. Additionally, note that, as reported in the manuscript, particularly in the conclusions, considering the only existing study on the encapsulation of pyrazole derive with solubilizing purposes and the only existing comparable results, the results obtained by us with both our nanotechnological approaches were unequivocally better and consequently appealing and important.

 Authors used G5k as a carrier for solubilizing pyrazole, while the safety of G5K is concerning. And it is very confusing to use paclitaxel as controls because paclitaxel is commonly regarded as a highly cytotoxic drug. Slightly lower cytotoxicity than paclitaxel does not mean G5k is a safe drug carrier.

Concerning the safety of G5K NPs, we disagree with the Reviewer. In fact, in addition to be remarkably and not “slightly” less cytotoxic than Paclitaxel (LD50 = 64.4 vs. 22.4 µM), used as positive control in the original version of the manuscript, the Reviewer must consider the intended use of the CR232-G5K NPs manufactured with G5K NPs, as well explained in the manuscript in lines 617-622 (revised version), reported below.

Moreover, considering the dose of CR232 which was active against SKMEL28 and HeLa cells (10 µM) [22], and considering that to release in vitro 10 µM of CR232 it should be necessary G5K only 0.24 µM, we can unequivocally assume that in a possible clinical application of a CR232 formulation based on G5K, G5K NPs will serve only as reservoir, protector and solubility enhancer for CR232, devoid of additional and unpleasant toxic effects, since at G5K 0.24 µM, the percentage of HeLa cell alive was 105 %.”

 According to these data, at the concentration necessary to provide an active dosage of CR232 as antiproliferative agent against SKMEL28 and HeLa cells, G5K is safe and well tolerated by cells.

In a word, this manuscript in this version is too premature to be published in this journal.

We are confident, that our manuscript, once revised according to the suggestions of all the Reviewers, is mature to be published on Nanomaterials.

Reviewer 3 Report

see attachment

Author Response

Manuscript “Successful Dendrimer and Liposomes-Based Strategies to Solubilize an Antiproliferative Pyrazole Otherwise Not Clinically Applicable“ by S. Alfei et al. presents huge amount of data on development of nanocarriers of anticancer drugs. Here authors synthesized and characterized by different methods two carrier systems dendrimers and SUVs to improve water solubility of Pyrazole. It is very important task and group has a lot of experience.

We are very thankful to the Reviewer for considering us as a group possessing a lot of experience in this field.

On my mind, this amount of presented data should be significantly better organized and analysed taking into account two important papers in the field. See below.

We are very thankful to the Reviewer for his advice. The suggested papers have been considered and read carefully finding interesting information. Accordingly, the introduction was improved by adding information from these works, which were then cited in the revised version of the manuscript. Please see lines 108-115. Concerning the organization of the work, also according to another Reviewer, the manuscript has been completely re-organized.

I am sure that authors should separate results and discussion.

Since it is allowed by Nanomaterials, following the extensive re-organization of the paper, we thought it was no longer necessary to separate the results from the discussion. We therefore ask the Reviewer to accept our decision.

Also, it is not clear why for vesicles (almost single nanocariers in clinical use) authors expected effect to increase of water solubility of drug. It is well-known that vesicles only increase circulation time of hydrophilic drugs.

We apologize in advance to the Reviewer, but we disagree with his statement. In fact, in the last 10 years the liposomal delivery systems have drawn attention as one of the noteworthy approaches to increase dissolution and subsequently absorption in the gastrointestinal (GI) tract of water-insoluble drugs. Please, consider:

Lee M. K. (2020). Liposomes for Enhanced Bioavailability of Water-Insoluble Drugs: In Vivo Evidence and Recent Approaches. Pharmaceutics12(3), 264. https://doi.org/10.3390/pharmaceutics12030264.

In this regard, the results obtained by us in the present study, confirmed the efficiency of liposomes in enhancing the water-solubility of drugs poorly soluble in water. Indeed, if the Reviewer considers Table 5, CR232-SUVs 30/1 was 1764.4-fold more soluble in water than the pristine CR232. 

Round 2

Reviewer 1 Report

Thank's to the authors as the manuscript gained clearly in clarity with the modifications made by them. There are still a few spelling/typo errors ( for example line 493 spactra por instead of spectra por)

Author Response

Thank's to the authors as the manuscript gained clearly in clarity with the modifications made by them. There are still a few spelling/typo errors (for example line 493 spactra por instead of spectra por)

We thank again the Reviewer for his previous suggestions which allowed us to considerably improve the form of our manuscript, and now we thank him for his current comment in which he appreciates the work done and approves the resulting manuscript. The signalled typo has been corrected, and similarly a few other oversights have been removed.

It seems to us that now the work is completely cleaned of any spelling error.

Reviewer 2 Report

It has been widely acknowledged that there are many pharmaceutical nanocarriers that can be utilized to enhance API with poor water solubility including pyrazole, such as micelles, nanoparticles, solid dispersions, nanocrystals, liposomes, etc. Here, authors reported their results by utilizing dendrimers and liposomes. What is the superiority of these two nanocarriers over others? Safety or drug loading? I believe commercial DSPE-PEG, mPEG-PDLLA micelles and Pluronic can be better pharmaceutical excipients than dendrimers. As a biocompatible and safe drug delivery system, liposomes have been widely applied to enhance water solubility of poor-soluble drugs (e.g., paclitaxel, docetaxel) and water-soluble drugs (e.g., doxorubicin) over decades of years. Replacing the reported drugs with pyrazole does not mean novelty. Do authors improve the formulations of liposomes to further enhance drug loading or biocompatibility or alter the biodistribution of pyrazole?

Besides, I have found that authors have published several similar papers in recent one year, such as utilizing dendrimes to enhance the water solubility of pyrazole and ursolic Acid (Biomedicines 2022, 10(1), 17; Nanomaterials 2021, 11(9), 2196; Nanomaterials 2021, 11(10), 2662). I am very confusing about the difference between these papers and this manuscript.

Figures and tables displayed here are hard to read, it is strongly suggested that authors read some good papers recently published in this journal to learn how to improve the readability of their manuscript (Nanomaterials 2022, 12, 153; Nanomaterials 2022, 12, 154).

Although authors have made some revisions of this manuscript, it is still very hard for me to recommend its acceptance in this journal.

Finally, Happy New Year to authors, editors and other reviewers for this manuscript. And I wish authors great achievements in next year.

Author Response

It has been widely acknowledged that there are many pharmaceutical nanocarriers that can be utilized to enhance API with poor water solubility including pyrazole, such as micelles, nanoparticles, solid dispersions, nanocrystals, liposomes, etc. Here, authors reported their results by utilizing dendrimers and liposomes. What is the superiority of these two nanocarriers over others? Safety or drug loading?

We respect the Reviewer’s opinion, concerning the existence of many pharmaceutical carriers such as micelles, nanoparticles, solid dispersions, nanocrystals, liposomes, etc. which were used to enhance the water-solubility of pyrazoles derivatives. Anyway, we are confident that, currently, considering also the 2022 literature, developed after the first submission of the present work, only one articles regarded the encapsulation of a pyrazole derivative in synthetic polymers with water-solubilizing purposes (Ref. 36), while no article exists regarding the solubilization in water of pyrazole derivatives using liposomes.

In this regard, when the first submission of our study occurred, the only results concerning the drug loading and the water solubility achieved by encapsulating a pyrazole derivative in commercial polymers, available to make comparisons with our ones, were those reported by Sun et al. (Ref.36). So, to answer the Reviewer’s question, concerning the superiority of our results over those achieved using different carrier, we can assert that the pyrazole formulations developed by us using dendrimer and liposomes were superior both for DL% and for the water-solubility. In fact, the micelles obtained by Sun proved a very low DL% (DL = 1.28%) and an insignificant (6.4 × 10-4 mg/mL) water solubility. In addition, concerning the dendrimer formulation, also the drug release profile was superior to that determined for the micelles developed by Sun (96% vs. 60%). Additionally, if the Reviewer wants also to consider the study now published on Journal of Applied Polymer Science, Volume 139, Issue 315, January 2022, by Costa R.D.O. et al., not yet published at the time of the first submission of our manuscript, both of our formulations are again superior for both DL% and encapsulation efficiency (EE%).

We hope to have satisfied the curiosity of the Reviewer.

I believe commercial DSPE-PEG, mPEG-PDLLA micelles and Pluronic can be better pharmaceutical excipients than dendrimers.

We respect the Reviewer’s opinion that commercial DSPE-PEG, mPEG-PDLLA micelles and Pluronic could be better pharmaceutical excipients than dendrimers, but there is no study to confirm this assertion for pyrazole derivatives. Moreover, while most of Pluronic NPs have displayed acute and subacute toxicity in mice (Micro & Nano Letters, 2013, Vol. 8, Iss. 11, pp. 796–800 doi: 10.1049/mnl.2013.0461), the results obtained by us using dendrimers are unequivocally superior to that obtained by other authors using commercial PEG-PLGA, PLA, DTC, and PVA, both in terms of DL%, EE%, water-solubility, and drug release profile.

As a biocompatible and safe drug delivery system, liposomes have been widely applied to enhance water solubility of poor-soluble drugs (e.g., paclitaxel, docetaxel) and water-soluble drugs (e.g., doxorubicin) over decades of years. Replacing the reported drugs with pyrazole does not mean novelty. Do authors improve the formulations of liposomes to further enhance drug loading or biocompatibility or alter the biodistribution of pyrazole?

We agree with the Reviewer that liposomes have been employed over decades of years to enhance the water solubility of several already marketed drugs, but not of molecules containing the pyrazole nucleus. In this regard, we were pioneers in having successfully evaluated the feasibility of solubilizing in water an insoluble bioactive pyrazole using liposomes. In addition to its novelty and originality, the Reviewer should recognize to our study the overall merit of having opened a new line of research that uses liposomes to improve the physicochemical characteristics of bioactive pyrazole derivatives. Although the DL% is already superior to those reported using other nanotechnological approaches (also optimized by using a simplex-centroid design as in Journal of Applied Polymer Science, Volume 139, Issue 315, January 2022, by Costa R.D.O. et al), we think that a further improvement could be always possible. 

Besides, I have found that authors have published several similar papers in recent one year, such as utilizing dendrimes to enhance the water solubility of pyrazole and ursolic Acid (Biomedicines 2022, 10(1), 17; Nanomaterials 2021, 11(9), 2196; Nanomaterials 2021, 11(10), 2662). I am very confusing about the difference between these papers and this manuscript.

We appreciate the interest of the Reviewer concerning our scientific production. Anyway, we make note that Biomedicines 2022, does not regard the encapsulation of the pyrazole derivative BBB4 in a dendrimer with solubilizing purposes, but the evaluation of the antibacterial potency of the BBB4-G4K NPs against different species of MDR Staphylococci. Concerning the other two papers the differences consist in the type of dendrimer and of encapsulated drugs and in the use of liposomes.

Figures and tables displayed here are hard to read, it is strongly suggested that authors read some good papers recently published in this journal to learn how to improve the readability of their manuscript (Nanomaterials 2022, 12, 153; Nanomaterials 2022, 12, 154).

We apologise in advance to the Reviewer, but we do not understand what is hard to read in the Figures and Tables reported in our manuscript. Both Tables and Figures respect the template and the instruction for authors provided by Nanomaterials. Moreover, as you have noted in the previous comment, we have published other articles on Nanomaterials (five in two years) reporting Figures and Tables presented with the same style and accepted as valid both by the Reviewers and by the journal. This Reviewer is the only one to disapprove our Figures and Tables’ style.

Although authors have made some revisions of this manuscript, it is still very hard for me to recommend its acceptance in this journal.

Fortunately, this Reviewer is the only one which finds hard to recommend the acceptance of our manuscript.

Finally, Happy New Year to authors, editors and other reviewers for this manuscript. And I wish authors great achievements in next year.

We thank the Reviewer for his good wishes and hoping for his greater availability, we reciprocate the wishes for a Happy New Year.

Reviewer 3 Report

Authors, should provide also clean version of manuscript. After so much changes in track change modus it is impossible to make revision. There are still a lot of mistakes in Eq number.

Author Response

Authors, should provide also clean version of manuscript. After so much changes in track change modus it is impossible to make revision. There are still a lot of mistakes in Eq number.

The Reviewer's request leaves us perplexed because in the submission phase we have sent both the version of the manuscript with highlighted the changed parts according to the opinion of this Reviewer and of Reviewer 2, but already cleaned up as regards the requests of Reviewer 1 (1 docx file and 1 pdf file), and the fully cleaned version of the manuscript both as docx file and as pdf file. We do not know what the Reviewer have received and downloaded, but we assure him that what he wants has been already provided.

We have carefully checked the equations number, which are correct.